# Mixture of Spectral Wavelets on Simplicial Complex: Analysis of Brain Connectome with Neurodegeneration

## Abstract

Understanding how pathological changes disrupt communication patterns in the brain requires models that examine interactions across multiple structural levels of a network. Many existing graph-learning methods emphasize node representations while providing limited treatment of signals defined on edges, and their use of predetermined spectral filters can restrict sensitivity to heterogeneous frequency behavior. To overcome these issues, we introduce a framework that couples a simplicial wavelet–based representation—capable of handling signals on both vertices and connections—with an adaptive filtering module that selects informative spectral components in a data-dependent manner. This combination enables flexible multi-scale analysis and highlights structural patterns relevant to neurodegenerative conditions. Evaluations on widely used brain graph benchmarks show consistent gains in predictive performance as well as clearer interpretation of disease-related network alterations. The implementation of this work will be released upon publication.

## 1 Introduction

Brain connectome provides a comprehensive map of the neural connections by connecting different anatomical regions of interest (ROIs) in the brain (Bullmore & Bassett, 2011). It offers an informative feature for understanding neurodegenerative diseases such as Alzheimer's Disease (AD) and Parkinson's Disease (PD) as they progressively impair brain function by disrupting neural connections (Pievani et al., 2014). The connectome defines a brain network which is represented as a graph, where nodes are ROIs and edges define connection strengths derived from neuroimaging modalities such as Diffusion Weighted Image (DWI) and functional Magnetic Resonance Imaging (fMRI). While analyzing these graphs helps identify disease-specific variations and supports early diagnosis and monitoring (Tijms et al., 2013), it often remains challenging due to the complex structure of brain networks and the inherent variability across individuals.

Recently, Graph Neural Network (GNN) and its variants have gained significant attention in brain network analysis (Li et al., 2021; Kan et al., 2022). While many of them operate in the naive graph space (Li et al., 2021), some utilize the spectral space of brain networks to address the heterogeneity of individual graphs, offering a powerful framework for modeling intricate topological structures and capturing underlying spectral properties (He et al., 2020). They leverage graph Fourier transform and filtering in the graph Fourier space to analyze signals on the graph, enabling the identification of critical regions associated with disease progression. The choice of filters and their bandwidths selectively captures global and localized features across different frequency components, highlighting disease-related alterations. Such approaches have shown promising results in identifying early biomarkers of neurodegenerative diseases, even under noise and inter-individual variability.

Most GNNs, both the spatial and spectral approaches, predominantly focus on node-level analyses, often treating edge features as auxiliary information within the adjacency matrix (Kipf & Welling, 2017; Gasteiger et al., 2019). While these approaches are effective for widely used graph benchmarks where edges represent simple categorical properties such as bond types or binary connectivity (Morris et al., 2020; Hu et al., 2020), their applicability is limited in brain networks. Edges in brain networks represent rich, continuous-valued connectomic features that capture the integrity of connections, providing critical insights into disease progression. While recent studies have introduced

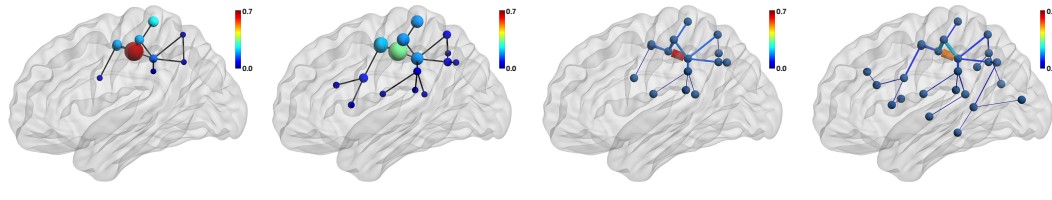

(a) Localized at a node with $s = 1$    (b) Localized at a node with $s = 4$    (c) Localized at an edge with $s = 1$    (d) Localized at an edge with $s = 4$

Figure 1: Visualization of diffusion filters (i.e., $\mathcal{K}_s(\lambda) = e^{-s\lambda}$) localized at a node (a and b) and an edge (c and d) in different scales. Smaller scales cover a narrow region, while larger scales enable broader diffusion, smoothing features across large neighborhood (low-pass filtering). Node size and color indicate node activation, while edge thickness and color represent edge activation.

the Hodge Laplacian to jointly analyze node and edge features (Yang et al., 2022), its application has been largely confined to the spatial domain, with little effort made to extend it to the spectral domain for a more comprehensive analysis.

Moreover, most existing spectral GNNs rely on fixed-bandwidth filtering, limiting their ability to capture diverse frequency components in graph-structured data. While polynomial approximations (Defferrard et al., 2016) and kernel-based methods (Xu et al., 2019) offer some flexibility, they typically apply uniform spectral filters across all nodes. This rigidity prevents models from adapting to regions that require different frequency responses—such as low-pass filtering for capturing global structures and band-pass filtering for preserving local variations—ultimately leading to suboptimal representations. More importantly, these methods lack mechanisms to dynamically adjust the filtering bandwidth and limits adaptability to diverse spectral patterns.

To address these limitations, we propose 1) Spectral Simplicial Wavelet Transform (SSWT) to derive multi-resolution node and edge features, and 2) Spectral Mixture of Experts (SpMoE) for adaptive multi-scale analysis. Leveraging the $r$-Laplacian (i.e., Hodge Laplacian), SSWT provides a richer multi-scale representation of brain networks explicitly on both nodes and edges, beyond traditional node-centric approaches. This framework decomposes a graph into local-to-global connectivity patterns of both nodes and edges at different resolutions with wavelets. Fig. 1 is the visualization of the filters designed as in the spectral space of $r$-Laplacians, where (a) and (b) are localized filters on nodes (i.e., $r$=0) and (c) and (d) are localized filters on edges (i.e., $r$=1). Furthermore, to improve adaptability to structural heterogeneity in spectral graph analysis, we introduce SpMoE, which dynamically selects relevant wavelet scales for downstream tasks. Inspired by Shazeer et al. (2017), SpMoE employs a gating mechanism to select spectral experts based on graph structure. By learning to prioritize informative scales for each graph, SpMoE adapts to the unique structural characteristics of individual brain networks and enhances generalizability across diverse neuroimaging datasets.

To this end, our key ideas introduce an integrated framework with the following **contributions**: **1)** We extend the Spectral Graph Wavelet Transform (SGWT) to higher order simplicial complex via Hodge Laplacian, enabling spectral analysis of both node and edge features in brain networks. **2)** We propose a Spectral Mixture of Experts (SpMoE) framework that combines multi-resolution spectral information during SSWT, capturing diverse scale resolutions for comprehensive graph representation. **3)** Our method enables graph-wise adaptive scale filtering, allowing dynamic and end-to-end learning of optimal scales tailored to each graph.

## 2    RELATED WORK

**Spectral Graph Neural Networks.** Spectral GNNs analyze graph-structured data by leveraging spectral representations grounded in graph signal processing (Shuman et al., 2013). Early spectral GNNs primarily relied on fixed-bandwidth filters to capture graph characteristics. ChebNet (Defferrard et al., 2016) introduced Chebyshev polynomial filters to approximate graph convolutions, while GCN (Kipf & Welling, 2017) further simplified this approach via first-order approximation. Diffusion-based approaches, such as GraphHeat (Xu et al., 2019) and GDC (Gasteiger et al., 2019), leverage the heat kernel to enhance low-pass properties in graph structures. Recent works introduced adaptive spectral filters with trainable parameters, enabling dynamic adjustment of frequency bandwidths based on the graph structure. GPR-GNN (Chien et al., 2021) refines polynomial filters through gradient-based optimization, and AdaGNN (Dong et al., 2021) employs channel-wise learn-

able parameters to capture significant frequencies. Furthermore, AGT (Cho et al., 2024) introduced a wavelet-based adaptive diffusion kernel to capture both low- and high-frequency components.

**Mixture of Experts.** The Mixture of Experts (MoE) framework enhances model performance by dividing a complex task into smaller, specialized sub-tasks (Jacobs et al., 1991; Jordan & Jacobs, 1994). Early MoE models used traditional machine learning techniques (Jordan et al., 1996; Chen et al., 1999), while modern variants incorporate sparse activation, where only a subset of experts is utilized for each input, significantly improving efficiency (Shazeer et al., 2017; Roller et al., 2021). It has been particularly successful in large-scale applications such as natural language processing (NLP), enabling the training of extremely large models (Fedus et al., 2022). Beyond NLP, MoE has also been adopted in computer vision (Riquelme et al., 2021; Zamfir et al., 2024), graph (Wang et al., 2023; Guo et al., 2025), and multi-modal learning (Shi et al., 2019; Mustafa et al., 2022).

## 3 PRELIMINARY

### 3.1 HODGE LAPLACIANS FOR GRAPHS

A simplicial complex is a structure that represents data as collections of simplices, which are fundamental building blocks such as points (0-simplex), lines (1-simplex), triangles (2-simplex), and their higher-dimensional counterparts. For any simplex included in a simplicial complex, all its faces (lower-dimensional simplices) must also be part of the complex. A graph can be viewed as a simplicial complex consisting of nodes (0-simplices) and edges (1-simplices), without higher-dimensional simplices ($r \geq 2$) (Schaub et al., 2020; Anand & Chung, 2023).

To formally describe relationships between $r$-simplices and $(r-1)$-simplices, the boundary operator $\partial_r : C_r \to C_{r-1}$ is introduced, where $C_r$ denotes the space of $r$-chains. An $r$-chain is a formal linear combination of $r$-simplices denoted as $\sigma^i$ expressed as $c = \sum_i \alpha_i \sigma^i$, where $\alpha_i$ is coefficient (i.e., 0 or 1). The boundary operator maps an $r$-simplex to its constituent $(r-1)$-simplices and is defined for an oriented $r$-simplex $\sigma^r = [v_0, v_1, \ldots, v_r]$ as:

$$\partial_r(\sigma^r) = \sum_{i=0}^{r} (-1)^i [v_0, v_1, \ldots, \hat{v}_i, \ldots, v_r], \tag{1}$$

where $\hat{v}_i$ denotes the omission of the $i$-th vertex. The boundary operator can be represented as a boundary matrix $B_r$, which relates $r$-simplices to $(r-1)$-simplices (e.g., edge-to-node connection). The elements of $B_r$ are defined as:

$$B_r(i, j) = \begin{cases} 1 & \text{if } \sigma_i^{r-1} \subset \sigma_j^r \text{ and orientations align,} \\ -1 & \text{if } \sigma_i^{r-1} \subset \sigma_j^r \text{ and orientations differ,} \\ 0 & \text{otherwise.} \end{cases} \tag{2}$$

Using the boundary matrix, the Hodge Laplacian for $r$-simplices is defined as $\mathcal{L}_r = B_r^T B_r + B_{r+1} B_{r+1}^T$. This definition extends the graph Laplacian to higher-order structures, enabling the analysis of multi-dimensional interactions (Dakurah et al., 2022). For graphs, where simplicial complexes consist of 0-simplices (nodes) and 1-simplices (edges), the boundary matrix $B_2 = \mathbf{0}$. Since $B_0 = \mathbf{0}$, as there are no lower-dimensional simplices, the Hodge Laplacians simplify to:

$$\mathcal{L}_0 = B_1 B_1^T, \quad \mathcal{L}_1 = B_1^T B_1. \tag{3}$$

Here, $B_1$ is the node-to-edge incidence matrix, which encodes the connectivity between nodes and edges in the graph. $\mathcal{L}_0$ is the graph Laplacian capturing node-level connectivity, while $\mathcal{L}_1$ characterizes edge-level relationships.

### 3.2 WAVELET TRANSFORM IN SIGNAL PROCESSING

Wavelet Transform is useful for analyzing signals across multiple resolutions, providing simultaneous localization in both time (or space) and frequency (Mallat, 1989). A mother wavelet $\psi$ (with a band-pass characteristic) on signal $x$ with scale $s$ and translation $\tau$ is defined as:

$$\psi_{s,\tau}(x) = \frac{1}{s} \psi \left( \frac{x - \tau}{s} \right), \tag{4}$$

where $s$ determines the dilation of the wavelet $\psi_{s,\tau}$. Small $s$ corresponds to higher frequencies (capturing fine details) and larger $s$ corresponds to lower frequencies (capturing global trends). The $\tau$ shifts the wavelet in space, enabling localized analysis of the signal. For a signal $f(x)$, a wavelet coefficient $W_f(s, \tau)$ is defined as:

$$W_f(s, \tau) = \langle f, \psi_{s,\tau} \rangle = \frac{1}{s} \int f(x) \psi^* \left( \frac{x - \tau}{s} \right) dx, \tag{5}$$

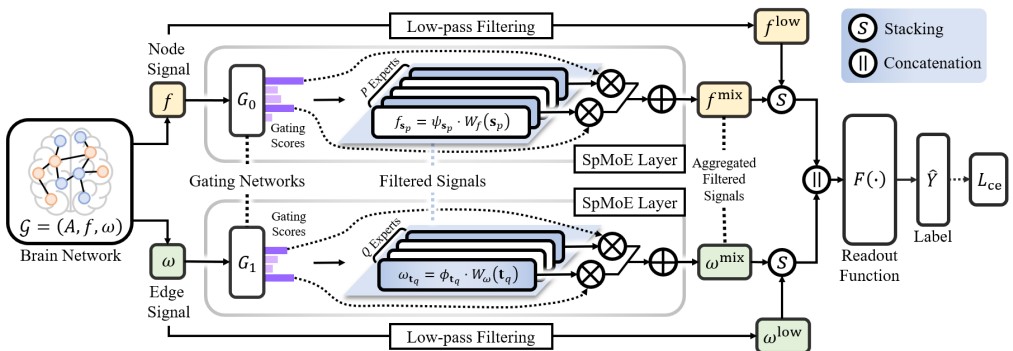

Figure 2: **Overall framework of our proposed method.** Given a brain network $\mathcal{G}$ with node signal $f$ and edge signal $\omega$, the signals pass through the gating networks ($G_0$ and $G_1$) which determine the appropriate scale experts to activate. The selected wavelet bases, $\psi_{\mathbf{s}_p}$ for nodes and $\phi_{\mathbf{t}_q}$ for edges, are then applied to generate the scale-filtered node and edge signals, $f_{\mathbf{s}_p}$ and $\omega_{\mathbf{t}_q}$, respectively. These filtered signals are aggregated based on their corresponding gating scores and combined with the low-pass filtered signals, $f^{\text{low}}$ and $\omega^{\text{low}}$. Finally, the concatenation of these components is fed into the readout function $F(\cdot)$ to predict the class label $\hat{Y}$.

where $^*$ denotes complex conjugate. It is invertible, allowing the original signal to be reconstructed:

$$f(x) = \frac{1}{C_\psi} \int \int W_f(s, \tau) \psi_{s,\tau}(x) d\tau \, ds, \tag{6}$$

where $C_\psi = \int \frac{|\Psi(j\omega)|^2}{|\omega|} d\omega < \infty$ is the admissibility constant with $\Psi$, i.e., the Fourier transform of the wavelet function $\psi$ in the frequency space $\omega$. An additional scaling function $\varphi$ is often introduced as a low-pass filter in addition to the mother wavelet to capture low-frequency components that mother wavelets cannot handle.

## 4 METHOD

Given a brain network represented by nodes corresponding to brain ROIs and edges representing their connections, along with associated node and edge signals, we aim to analyze the network by proposing a higher-order version of Spectral Graph Wavelet Transform (SGWT) (Hammond et al., 2011), as illustrated in Fig. 2. In Sec. 4.1, we extend the definition of SGWT with Hodge Laplacians, enabling the decomposition of node and edge signals at specific scales. To learn the optimal scale for each node and edge, Sec. 4.2 introduces adaptive wavelet bases, and a Spectral Mixture of Experts (SpMoE) for multi-resolution analysis is introduced in Sec. 4.3 to dynamically select the most relevant scales at each node and edge. Finally, Sec. 4.4 presents the training objective for a classification task by introducing load balancing losses to ensure equitable utilization of experts.

### 4.1 SPECTRAL SIMPLICIAL WAVELET TRANSFORM

Let $\mathcal{G} = (A, f, \omega)$ be a directed graph, where $A \in \mathbb{R}^{N \times N}$ is an adjacency matrix that defines the connectivity between $N$ nodes, $f \in \mathbb{R}^{N \times d_f}$ represents $d_f$-dimensional node signals and $\omega \in \mathbb{R}^{M \times d_\omega}$ denotes $d_\omega$-dimensional edge signals for $M$ edges. From $A$, the relationships between nodes and edges can be captured in the incidence matrix $B_1 \in \mathbb{R}^{N \times M}$, and using $B_1$, the Hodge Laplacians $\mathcal{L}_0$ and $\mathcal{L}_1$ are defined as in Eq. (3). Both $\mathcal{L}_0$ and $\mathcal{L}_1$ are real and positive semi-definite and can be decomposed into $\mathcal{L}_0 = U_0 \Lambda_0 U_0^T$ and $\mathcal{L}_1 = U_1 \Lambda_1 U_1^T$, where $U_0 = [u_{0,1}, \ldots, u_{0,N}]$ and $U_1 = [u_{1,1}, \ldots, u_{1,M}]$ are orthonormal matrices, and $\Lambda_0 = \text{diag}(\lambda_{0,1}, \ldots, \lambda_{0,N})$ and $\Lambda_1 = \text{diag}(\lambda_{1,1}, \ldots, \lambda_{1,M})$ are diagonal matrices containing the non-negative eigenvalues.

To analyze graph signals $f$ and $\omega$ at multiple scales, we extend the SGWT (Hammond et al., 2011) to Spectral Simplicial Wavelet Transform (SSWT). SSWT enables the decomposition of multi-order graph signals into multi-scale components across various scales. The wavelet bases for nodes and edges are defined as:

$$\psi_s = U_0 \mathcal{K}(s\Lambda_0) U_0^T, \quad \phi_t = U_1 \mathcal{K}(t\Lambda_1) U_1^T, \tag{7}$$

where $\mathcal{K}$ is a kernel function and $s, t \in \mathbb{R}$ are scale parameters. These basis functions (also visualized in Fig. 1) are localized in the graph space, capturing features specific to individual nodes ($\delta_n$) or edges ($\delta_m$) at a fixed resolution. The graph signals $f$ and $\omega$ are projected onto the spectral domain using these bases, yielding wavelet coefficients $W_f(s) = \psi_s \cdot f$ and $W_\omega(t) = \phi_t \cdot \omega$. Under the admissibility condition, the signals can be perfectly reconstructed via the Inverse Simplicial Wavelet

Transform (ISWT):

$$f = \frac{1}{C_{\mathcal{K}}} \int_0^\infty \psi_s \cdot W_f(s) \frac{ds}{s}, \quad \omega = \frac{1}{C_{\mathcal{K}}} \int_0^\infty \phi_t \cdot W_\omega(t) \frac{dt}{t}, \tag{8}$$

where $C_{\mathcal{K}} = \int_0^\infty \frac{\mathcal{K}^2(\lambda)}{\lambda} d\lambda < \infty$. Eq. (8) represents the superposition of multi-resolution representations of $f$ and $\omega$ over scales $s, t \in [0, \infty)$. Each component at scales $s$ and $t$ is given as:

$$f_s = \psi_s \cdot W_f(s) = U_0 \mathcal{K}^2(s\Lambda_0) U_0^T f, \quad \omega_t = \phi_t \cdot W_\omega(t) = U_1 \mathcal{K}^2(t\Lambda_1) U_1^T \omega. \tag{9}$$

This enables the extraction of information corresponding to the desired resolution of signals, allowing for detailed multi-resolution analysis in the spatial domain.

**Orientation Equivariance.** When a brain network is represented as an undirected graph, constructing $B_1$ requires assigning an orientation to each edge. While $\mathcal{L}_0$ is invariant to the choice of orientation, $\mathcal{L}_1$ depends on the sign of each element, i.e., edge. This dependency poses challenges in ensuring that the graph analysis results are consistent and robust to arbitrary orientation choices. To address this, we prove that our method satisfies *orientation equivariance*, a property inspired by simplicial neural networks (Schaub et al., 2020; Yang et al., 2022) where maintaining consistency across arbitrary edge orientations is essential. Specifically, orientation equivariance ensures that altering the relative orientations of certain edges results in sign flips in the output embeddings.

**Lemma 1.** *Consider a boundary matrix $B_1 \in \mathbb{R}^{N \times M}$ with arbitrary orientations and edge signal $\omega$ of a graph $\mathcal{G}$. Let $P \in \mathbb{R}^{M \times M}$ be an arbitrary diagonal matrix encoding changes in edge orientation, where $P[i,i] = -1$ if the orientation of the $i$-th edge is flipped, and $P[i,i] = 1$ otherwise. Define the reoriented graph $\mathcal{G}'$, with $\omega' = P\omega$ and $B_1' = B_1 P$. Then, the filtered edge signal $\omega_t'$ on $\mathcal{G}'$ satisfies $\omega_t' = P\omega_t$, demonstrating that the ISWT is equivariant to edge orientation.*

Lemma 1 tells that flipping the orientation of certain edges changes the sign of the corresponding components of $\omega_t$, while preserving their magnitudes as in the proof in Sec. A. This ensures that $\mathcal{L}_1$ can be reliably utilized for robust graph analysis in any undirected setting.

## 4.2 ADAPTIVE SCALING FOR NODES AND EDGES

Conventional graph convolution aggregates information uniformly across nodes with respect to the number of hidden layers, which often resulting in over-smoothing or under-smoothing, particularly when there are significant local variations. To address this, we adopt an adaptive scaling approach for both nodes and edges, allowing them to aggregate information optimally based on their local graph structure. We define node-wise and edge-wise wavelet bases using trainable set of scaling parameters, $\mathbf{s} = \{s_i\}_{i=1}^N$ associated with nodes $\{n_i\}_{i=1}^N$ and $\mathbf{t} = \{t_i\}_{i=1}^M$ associated with edges $\{e_i\}_{i=1}^M$. The adaptive wavelet bases $\psi_{s_i}$ at $n_j$ localized at $n_i$ and $\phi_{t_i}$ at $e_j$ localized at $e_i$ are formulated as:

$$\psi_{s_i, n_i}(n_j) = \sum_{\ell=1}^N \mathcal{K}(s_i \lambda_{0,\ell}) u_{0,\ell}^*(n_i) u_{0,\ell}(n_j), \quad \phi_{t_i, e_i}(e_j) = \sum_{\ell=1}^M \mathcal{K}(t_i \lambda_{1,\ell}) u_{1,\ell}^*(e_i) u_{1,\ell}(e_j). \tag{10}$$

By adjusting the scales $s_i$ and $t_i$, this approach enhances or suppresses specific details in the spatial domain as needed, enabling a more nuanced analysis of graph signals.

## 4.3 SPECTRAL MIXTURE OF EXPERTS

The key to effectively analyzing brain networks lies in understanding their spectral multi-scale characteristics, which play a crucial role in capturing both global and local graph patterns. We propose SpMoE framework that dynamically combines spectral information across multiple scales. Inspired by Shazeer et al. (2017), we define multiple learnable sets of wavelet scales for nodes and edges, treating each set as an expert. We further design a mechanism to adaptively select the optimal scale based on the graph's structural features. The filtering results at these wavelet scales are combined to generate multi-resolution graph features, capturing diverse scale-specific characteristics.

**Gating network.** To dynamically determine the most appropriate scale sets for nodes and edges in each graph, we employ a noisy top-$k$ gating mechanism (Shazeer et al., 2017) as the gating network in our SpMoE layer. This gating mechanism adaptively selects the most informative spectral experts, ensuring that only the most relevant wavelet scale sets contribute to the final representation. This is achieved using the `TopK` function, which retains the top-$k$ gating scores while setting the remaining scores to $-\infty$. Formally, given the input graph signals $x = \{f, \omega\}$ and $k \in \mathbb{R}$, the gating function $G$ is defined as:

$$G(x) = \text{Softmax}(\text{TopK}(H(x), k)), \tag{11}$$

$$H(x)_i = (x \cdot \theta_g)_i + \epsilon \cdot \text{Softplus}((x \cdot \theta_{\text{noise}})_i), \tag{12}$$

where $H(x)_i$ represents raw gating score for $i$-th expert, $\epsilon$ denotes standard Gaussian noise, and $\theta_g$ and $\theta_{\text{noise}}$ are learnable weight matrix for computing gating scores and the noise amplitude, respectively. In Eq. (12), the $\epsilon$, whose amplitude is controlled by $\theta_{\text{noise}}$, is introduced to enhance robustness and ensure balanced activation of scales during training, preventing over-reliance on specific experts. The detailed role of $\theta_{\text{noise}}$ are further explained in Sec. B.

**SpMoE layer.** The SpMoE layer aggregates multi-scale spectral features for both nodes and edges, enabling the model to adaptively select and combine the most relevant scales for each graph. Using the gating functions defined in Eq. (11), the SpMoE layer dynamically determines the importance of each scale and combines the outputs of the corresponding experts. With $P$ experts for nodes and $Q$ experts for edges, the aggregated node and edge representations are computed as:

$$f^{\text{mix}} = \sigma(\sum_{p=1}^{P} G_0(f)_p f_{\mathbf{s}_p} \theta_0), \quad \omega^{\text{mix}} = \sigma(\sum_{q=1}^{Q} G_1(\omega)_q \omega_{\mathbf{t}_q} \theta_1), \tag{13}$$

where $f_{\mathbf{s}_p} \in \mathbb{R}^{N \times d_f}$ and $\omega_{\mathbf{t}_q} \in \mathbb{R}^{M \times d_\omega}$ represent features filtered at scale set $\mathbf{s}_p = \{s_{p,i}\}_{i=1}^{N}$ and $\mathbf{t}_q = \{t_{q,i}\}_{i=1}^{M}$ according to Eq. (9), and $p$ and $q$ are indices for node and edge experts. $\sigma(\cdot)$ denotes an activation function, and $\theta_0 \in \mathbb{R}^{d_f \times h}$ and $\theta_1 \in \mathbb{R}^{d_\omega \times h}$ are learnable weight matrices shared across all scales in the $h$-dimensional space. The gating functions $G_0$ and $G_1$ assign weights to the filtered features, dynamically selecting the most relevant scales for the graph. When the gating value is 0, the corresponding expert is not activated, and its computation is skipped and reduces computational overhead. By adaptively combining features across scales, the SpMoE layer effectively captures diverse structural properties of the graph, offering adaptability to heterogeneous graph structures.

**Choice of Kernels.** The choice of wavelet kernel determines the graph characteristics to be captured. For low-pass filtering, we adopt the kernel $\mathcal{K}_s^{\text{low}}(\lambda) = e^{-s\lambda}$ as a scaling function, which smooths local variations and emphasizes global structures by aggregating information over a broad range of nodes and edges. To capture band-pass characteristics, the kernel $\mathcal{K}_s(\lambda) = s\lambda e^{-s\lambda}$ is employed, emphasizing intermediate frequencies and highlighting localized features in the graph.

In low-pass filtering, the scales $\mathbf{s}^{\text{low}}$ for nodes and $\mathbf{t}^{\text{low}}$ for edges are consistently learned and applied across the graph. The filtered representations are computed as $f^{\text{low}} = \sigma(f_{\mathbf{s}^{\text{low}}} \theta_0)$ and $\omega^{\text{low}} = \sigma(\omega_{\mathbf{t}^{\text{low}}} \theta_1)$, where $f_{\mathbf{s}^{\text{low}}} \in \mathbb{R}^{N \times d_f}$ and $\omega_{\mathbf{t}^{\text{low}}} \in \mathbb{R}^{M \times d_\omega}$ represent node and edge features filtered using the low-pass kernel according to Eq. (9). For band-pass filtering, multiple scales are adaptively combined using SpMoE layer, as defined in Eq. (13). Finally, the outputs of low-pass and band-pass filtering are stacked to form the final node and edge features as $\tilde{f} = [f^{\text{low}}, f^{\text{mix}}] \in \mathbb{R}^{N \times 2h}$ and $\tilde{\omega} = [\omega^{\text{low}}, \omega^{\text{mix}}] \in \mathbb{R}^{M \times 2h}$. These embeddings balance global and localized information to improve downstream tasks.

### 4.4 TRAINING OBJECTIVE WITH LOAD BALANCING

**Load balancing losses.** To ensure balanced use of experts during training in the SpMoE layer, we incorporate two complementary losses as in Shazeer et al. (2017), i.e., importance loss and load loss. These losses prevent the model from over-relying on specific experts.

The importance loss enforces uniform gating activations across all experts, ensuring balanced utilization during training. For a batch of graph signals $X$, where each signal is either $f$ or $\omega$, the importance of the $i$-th expert is computed as the sum of gating activations across the batch:

$$\text{Imp}_i(X) = \sum_{x \in X} G_i(x), \tag{14}$$

where $G_i(x)$ is the gating value assigned to the $i$-th expert for graph signal $x$. To address imbalances in importance across experts for both node and edge, the importance loss $L_{\text{imp}}$ is formulated as:

$$L_{\text{imp}}(f, \omega) = CV(\text{Imp}(f))^2 + CV(\text{Imp}(\omega))^2, \tag{15}$$

where $CV(\cdot)$ denotes the coefficient of variation.

While the importance loss ensures balanced activation levels across experts, it does not guarantee that graphs are distributed fairly among them. In particular, an expert might be activated frequently with low gating values, resulting in a limited contribution to the training process and imbalanced workloads. To address this issue, we introduce the load loss which focuses on balancing the number

of graphs assigned to each expert during training. The load for expert $i$ over batch $X$ is defined as the total number of graphs for which the expert is activated:

$$\text{load}_i(X) = \sum_{x \in X} \mathbb{P}(G(x)_i \neq 0), \tag{16}$$

where $\mathbb{P}(G(x)_i \neq 0)$ is the probability of activating $i$-th expert for a given graph signal $x$. This probability is influenced by the noise term in Eq. (12), which introduces stochasticity in expert selection. A detailed derivation of $\mathbb{P}(G(x)_i \neq 0)$ is provided in Sec. B. To penalize variations in graph assignments across experts for both node and edge signals, the load loss is formulated as:

$$L_{\text{load}}(f, \omega) = CV(\text{load}(f))^2 + CV(\text{load}(\omega))^2. \tag{17}$$

**Training Objective.** The goal of the model is to predict the true graph label $Y$ as $\hat{Y}$. To achieve this, the combined graph embedding $[\tilde{f}; \tilde{\omega}] \in \mathbb{R}^{(N+M) \times 2h}$, obtained by concatenating the node embeddings $\tilde{f}$ and edge embeddings $\tilde{\omega}$, is passed through a readout function $F(\cdot)$ (i.e., MLP with activation functions). Using cross-entropy, the loss for the prediction is defined as:

$$L_{\text{ce}} = \sum_{b=1}^{\mathcal{B}} \sum_{c \in C} Y_{bc} \cdot \log(\hat{Y}_{bc}), \tag{18}$$

where $Y_{bc}$ represents the true class label for the $b$-th sample and the $c$-th class, and $\hat{Y}_{bc}$ denotes the predicted probability of the corresponding class. The overall training objective integrates balancing terms from the SpMoE layer to encourage efficient utilization of all experts, as described earlier:

$$L = L_{\text{ce}} + \alpha\big(L_{\text{load}} + L_{\text{imp}}\big), \tag{19}$$

where $\alpha$ adjusts the influence of the load balancing losses. The Eq. (19) not only optimizes prediction accuracy but also ensures fair distribution of computations across experts.

## 5 EXPERIMENT

In this section, we provide the experimental setup, quantitative comparisons with baseline models, and ablation studies. We then discuss clinical interpretation and class-specific expert selection.

### 5.1 DATASET AND EXPERIMENT SETTINGS

Two brain network datasets are used: 1) the Alzheimer's Disease Neuroimaging Initiative (ADNI) and 2) the Parkinson's Progression Markers Initiative (PPMI). Diffusion MRI in ADNI was used to construct structural connectomes, as they capture anatomical degeneration (Vemuri & Jack, 2010)—such as white matter disruption and cortical atrophy, and AD pathology is known to propagate along these structural pathways (Raj et al., 2012). To further assess the generalizability of our model, we also evaluate it on the functional connectivity from PPMI. Detailed demographics and graph construction are given in Sec. C.

**ADNI Dataset.** The ADNI study (Mueller et al., 2005) is a public repository for AD research that provides multi-modal imaging and biomarkers at various stages of cognitive decline. Structural brain networks are constructed using probabilistic tractography on DWI, with 160 brain regions from Destrieux atlas (Destrieux et al., 2010) as nodes, and white-matter fiber connections weighted by tract counts as edges. Node features include region-wise cortical thickness (CT) from MRI and Standardized Uptake Value Ratios (SUVRs) from FDG-PET. Subjects are grouped into five categories: Cognitively Normal (CN), Significant Memory Concern (SMC), Early Mild Cognitive Impairment (EMCI), Late MCI (LMCI), and AD.

**PPMI Dataset.** PPMI (Marek et al., 2011) provides imaging and biomarkers for PD progression. Functional brain networks are derived from fMRI on 116 regions based on the AAL atlas (Tzourio-Mazoyer et al., 2002), with Blood-Oxygen-Level-Dependent (BOLD) signals as node features and functional connectivity (correlation) between regions as edge features (Xu et al., 2023). The dataset contains three diagnostic groups: CN, Prodromal, and PD.

**Implementation.** Since utilizing $\mathcal{L}_1 \in \mathbb{R}^{M \times M}$ introduces substantial computational overhead as $M$ increases, we averaged all brain networks and removed edges below a predefined threshold to ensure computational feasibility and consistency. Regarding computational cost, we present an analysis of runtime and memory usage in Sec. E. Further implementation details, including expert configurations and hyperparameter settings for both our model and baselines, are provided in Sec. D.

Table 1: Classification performance on the ADNI and PPMI datasets evaluated using 5-fold cross-validation. The best results are shown in **bold**, and the second-best results are underlined.

| Model | ADNI-CT | | | ADNI-FDG | | | PPMI | | |
|---|---|---|---|---|---|---|---|---|---|
| | Accuracy (%) | Precision (%) | Recall (%) | Accuracy (%) | Precision (%) | Recall (%) | Accuracy (%) | Precision (%) | Recall (%) |
| SVM (Linear) | 82.4 ± 2.7 | 82.2 ± 3.3 | 85.2 ± 2.5 | 85.3 ± 2.1 | 85.7 ± 2.7 | 86.9 ± 2.1 | 60.5 ± 10.1 | 30.2 ± 6.6 | 28.0 ± 8.2 |
| MLP (2-layers) | 78.8 ± 2.2 | 79.2 ± 3.6 | 79.9 ± 2.6 | 87.5 ± 1.6 | 88.2 ± 2.4 | 88.1 ± 1.4 | 68.9 ± 3.5 | 36.3 ± 4.4 | 39.0 ± 8.6 |
| GCN | 61.4 ± 3.1 | 59.8 ± 2.5 | 62.6 ± 4.4 | 68.8 ± 2.0 | 67.7 ± 2.8 | 69.7 ± 2.5 | 78.8 ± 2.1 | 48.1 ± 5.2 | 70.3 ± 4.3 |
| GAT | 64.2 ± 5.5 | 62.7 ± 6.7 | 66.8 ± 4.6 | 69.2 ± 7.1 | 67.0 ± 10.6 | 73.6 ± 3.7 | 81.2 ± 2.4 | 51.4 ± 6.9 | 77.2 ± 5.5 |
| GDC | 77.1 ± 4.3 | 76.9 ± 5.0 | 78.5 ± 4.4 | 86.2 ± 3.2 | 86.7 ± 3.3 | 87.0 ± 2.9 | 73.0 ± 0.7 | 36.5 ± 3.1 | 61.8 ± 9.4 |
| GraphHeat | 70.9 ± 3.2 | 70.3 ± 3.0 | 71.8 ± 2.6 | 77.0 ± 2.4 | 77.5 ± 3.5 | 77.3 ± 1.0 | 79.1 ± 2.0 | 48.4 ± 4.7 | 84.5 ± 3.0 |
| ADC | 82.1 ± 2.4 | 77.6 ± 1.9 | 72.8 ± 6.7 | 88.6 ± 2.8 | 70.8 ± 6.2 | 75.3 ± 5.3 | 78.8 ± 2.3 | 50.7 ± 8.0 | 66.9 ± 5.5 |
| BrainGNN | 69.3 ± 2.8 | 20.1 ± 0.5 | 23.4 ± 3.8 | 68.9 ± 2.4 | 20.3 ± 0.4 | 31.9 ± 13.3 | 69.6 ± 5.1 | 38.5 ± 7.9 | 70.5 ± 5.0 |
| BrainNetTF | 87.8 ± 3.9 | 65.9 ± 16.7 | 70.6 ± 9.0 | 87.1 ± 5.3 | 65.5 ± 16.5 | 66.3 ± 15.6 | 71.3 ± 4.1 | 42.6 ± 10.5 | 76.1 ± 6.3 |
| Exact | 86.2 ± 2.0 | 86.6 ± 1.7 | 86.7 ± 2.3 | 90.2 ± 2.7 | 90.7 ± 2.8 | 90.7 ± 2.8 | 79.5 ± 2.4 | 48.1 ± 5.4 | 76.6 ± 9.0 |
| ALTER | 84.4 ± 2.2 | 68.3 ± 10.9 | 72.7 ± 11.2 | 85.1 ± 4.1 | 66.0 ± 12.5 | 67.8 ± 14.4 | 82.5 ± 6.3 | 57.8 ± 6.6 | 79.4 ± 4.7 |
| ContrastPool | 63.3 ± 4.1 | 61.7 ± 2.9 | 64.0 ± 4.5 | 75.3 ± 1.3 | 74.7 ± 1.3 | 76.2 ± 1.9 | 71.3 ± 5.2 | 47.0 ± 5.9 | 74.8 ± 11.9 |
| AGT | 90.3 ± 1.8 | 91.3 ± 2.4 | 89.9 ± 2.5 | 94.8 ± 1.1 | 94.3 ± 1.5 | 95.3 ± 1.4 | 83.6 ± 3.8 | 62.4 ± 4.4 | 87.6 ± 3.5 |
| SSWT w/o SpMoE (ours) | 92.5 ± 1.1 | 92.0 ± 1.1 | 93.4 ± 1.0 | 92.8 ± 0.7 | 93.3 ± 1.3 | 93.1 ± 0.9 | 84.6 ± 2.7 | 63.2 ± 5.5 | 85.1 ± 2.2 |
| SSWT w/ SpMoE (ours) | **94.9 ± 1.3** | **94.4 ± 1.7** | **95.4 ± 0.8** | **96.4 ± 0.9** | **96.5 ± 1.1** | **97.1 ± 1.1** | **87.4 ± 2.1** | **65.3 ± 3.4** | **88.3 ± 1.6** |

Table 2: Grad-CAM outcomes of the top-10 ROIs and connectomes with the highest activation (Act.) for AD/PD from the ADNI-CT (left) / PPMI (right) results. The indices given in Destrieux (Destrieux et al., 2010) and AAL (Tzourio-Mazoyer et al., 2002) atlas.

| ADNI-CT | | | | | | PPMI | | | | | |
|---|---|---|---|---|---|---|---|---|---|---|---|
| Idx | ROI (Node) | Act. | Idx1-Idx2 | ROI1-ROI2 (Edge) | Act. | Idx | ROI (Node) | Act. | Idx1-Idx2 | ROI1-ROI2 (Edge) | Act. |
| 158 | R-Thalamus_Proper | 0.977 | 153-154 | L-Putamen - L-Pallidum | 0.750 | 74 | Putamen_R | 0.717 | 74-78 | Putamen_R - Thalamus_R | 0.887 |
| 156 | R-Caudate | 0.867 | 159-160 | R-Putamen - R-Pallidum | 0.715 | 77 | Thalamus_L | 0.688 | 73-77 | Putamen_L - Thalamus_L | 0.815 |
| 160 | R-Pallidum | 0.866 | 99-147 | R-G_pariet_inf-Angular - R-S_temporal_sup | 0.453 | 73 | Putamen_L | 0.581 | 31-71 | Cingulum_Ant_L - Caudate_L | 0.801 |
| 154 | L-Pallidum | 0.820 | 38-73 | L-G_temporal_middle - L-S_temporal_sup | 0.412 | 20 | Supp_Motor_Area_R | 0.492 | 20-58 | Supp_Motor_Area_R - Postcentral_R | 0.698 |
| 159 | R-Putamen | 0.814 | 112-147 | R-G_temporal_middle - R-S_temporal_sup | 0.411 | 78 | Thalamus_R | 0.489 | 77-99 | Thalamus_L - Cerebellum_6_L | 0.662 |
| 152 | L-Thalamus_Proper | 0.745 | 29-45 | L-G_precentral - L-S_central | 0.393 | 11 | Frontal_Inf_Oper_L | 0.479 | 32-72 | Cingulum_Ant_R - Caudate_R | 0.634 |
| 151 | L-Hippocampus | 0.556 | 28-45 | L-G_postcentral - L-S_central | 0.379 | 76 | Pallidum_R | 0.444 | 19-57 | Supp_Motor_Area_L - Postcentral_L | 0.576 |
| 149 | L-Amygdala | 0.554 | 15-52 | L-G_front_middle - L-S_front_inf | 0.377 | 91 | Cerebellum_Crus1_L | 0.438 | 4-38 | Frontal_Sup_R - Hippocampus_R | 0.484 |
| 155 | R-Amygdala | 0.551 | 89-127 | R-G_front_middle - R-S_front_middle | 0.362 | 75 | Pallidum_L | 0.421 | 33-78 | Cingulum_Mid_L - Thalamus_R | 0.411 |
| 157 | R-Hippocampus | 0.550 | 103-119 | R-G_precentral - R-S_central | 0.347 | 70 | Paracentral_Lobule_R | 0.418 | 30-82 | Insula_R - Temporal_Sup_R | 0.400 |

**Evaluation.** For fair evaluation, we compared our method against a wide range of baselines, including recent state-of-the-art models. These include conventional classifiers (Linear SVM, 2-layer MLP); spatial GNNs (GCN (Kipf & Welling, 2017), GAT (Veličković et al., 2018)); spectral and diffusion-based GNNs (GDC (Gasteiger et al., 2019), GraphHeat (Xu et al., 2019), ADC (Zhao et al., 2021), Exact (Choi et al., 2022), AGT (Cho et al., 2024)); and brain-specific models (BrainGNN (Li et al., 2021), BrainNetTF (Kan et al., 2022), ALTER (Yu et al., 2024), ContrastPool (Xu et al., 2024)). We evaluated all models with 5-fold cross-validation to ensure unbiased results for accuracy, precision, and recall.

### 5.2 Quantitative results

**Comparison with baselines.** The results for ADNI and PPMI are summarized in Tab. 1. Ours (SSWT w/ SpMoE) consistently outperforms all baseline methods across datasets and metrics including recall—a critical indicator of sensitivity. While AGT performs best among the baselines across all metrics and datasets, our model brings improvements with average gains of 3.3%p in accuracy, 2.7%p in precision, and 2.7%p in recall across the three datasets. Notably, incorporating SpMoE into SSWT yields substantial gains over its non-expert variant, highlighting the effectiveness of our graph-wise adaptive multi-scale filtering. These results emphasize the robustness and generalizability of our model in capturing meaningful patterns in brain networks.

**Ablation Study.** To assess the effect of node and edge signals and different filter types in SSWT, we conduct an ablation study on the ADNI-CT (Tab. 3). Using both node and edge signals yields the highest accuracy, confirming their complemen-

Table 3: Ablation study on the SSWT for the ANDI-CT.

| Use of signals | | Use of filters | | Accuracy (%) | Precision (%) | Recall (%) |
|---|---|---|---|---|---|---|
| Node | Edge | Low pass | Band pass | | | |
| ✓ | ✗ | ✓ | ✓ | 91.8 ± 0.9 | 91.4 ± 1.0 | 93.1 ± 1.1 |
| ✗ | ✓ | ✓ | ✓ | 90.9 ± 1.5 | 91.0 ± 1.9 | 91.5 ± 1.5 |
| ✓ | ✓ | ✓ | ✓ | **94.9 ± 1.3** | **94.4 ± 1.7** | **95.4 ± 0.8** |
| ✓ | ✓ | ✗ | ✓ | 94.4 ± 1.3 | 94.0 ± 1.4 | 95.2 ± 1.1 |
| ✓ | ✓ | ✓ | ✗ | 92.3 ± 1.1 | 91.7 ± 1.4 | 93.3 ± 0.9 |

tary roles in capturing structural properties of brain networks. Removing either signal leads to a performance drop, emphasizing their contributions to regional characterization and connectivity. For filter selection, combining low-pass and band-pass filters achieves the best performance, balancing global and localized information. Excluding either filter reduces performance, highlighting their role in capturing fine-grained spectral features as well as global structural information. Additional ablation studies on the expert configurations and the load balancing losses are provided in Sec. E.

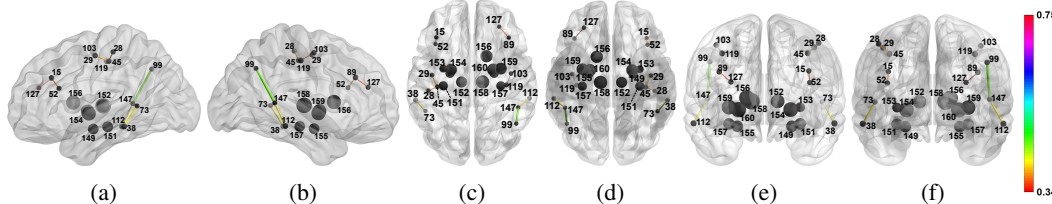

Figure 3: Top-10 activated ROIs and connectomes for ADNI-CT classification. (a)/(b): outer view of left/right hemisphere, (c)/(d): top/bottom, (e)/(f): front/rear view. Node size and edge color/thickness indicate activations, with node indices corresponding to the Destrieux atlas (Destrieux et al., 2010).

## 5.3 MODEL BEHAVIOR ANALYSIS

**Interpretation.** To provide clinical interpretability, we investigate the class-averaged Grad-CAM (Selvaraju et al., 2017) of nodes and edges in classifying AD and PD. Tab. 2 reports the top-10 ROIs and connectomes with the highest gradient activation for ADNI-CT and PPMI experiments.

For ADNI-CT, the results are visualized in Fig. 3. All highly activated nodes are located within subcortical structures (e.g., hippocampus, amygdala), which are crucial for memory, cognitive processing, and motor functions, and are strongly associated with AD (Tentolouris-Piperas et al., 2017; Shukla et al., 2024). The thalamus exhibits the highest activation, aligning with evidence that its atrophy correlates with cognitive decline (de Jong et al., 2008). The selected edges connect both subcortical and cortical regions, particularly in the temporal and frontal lobes, which play essential roles in memory, language, and executive functions (Palmer et al., 1987; Killiany et al., 1993).

For PPMI, the putamen, pallidum, and caudate-key structures of the basal ganglia-exhibited high activation in both nodes and edges across both hemispheres. These regions, essential for motor control, cognition, and emotion, are strongly linked to PD (de la Fuente-Fernández, 2013; Obeso et al., 2000). Additionally, strong activation is observed in the putamen-thalamus and thalamus-cerebellum connections, consistent with known PD-related disruptions in the basal ganglia-thalamic and cerebello-thalamo-cortical circuits (Helmich et al., 2012). Across both datasets, our model consistently captures bilaterally symmetric activation patterns, reinforcing its classification reliability.

**Class-Specific Expert Selection.** The SpMoE dynamically selects spectral experts based on the structural properties of each brain network. To analyze class-wise expert selection, we visualize the frequency of node expert activations across diagnostic groups in ADNI-CT, using five experts and $k = 2$ (Fig. 4). Although the load balancing losses ensure an overall even distribution of expert usage, selection patterns vary significantly across classes. Within each class, the model exhibits distinct activation preference, highlighting that the gating mechanism effectively differentiates neurodegeneration stages. These class-specific patterns indicate that SpMoE dynamically adjusts its scale selection to capture the most relevant spectral features for each disease stage.

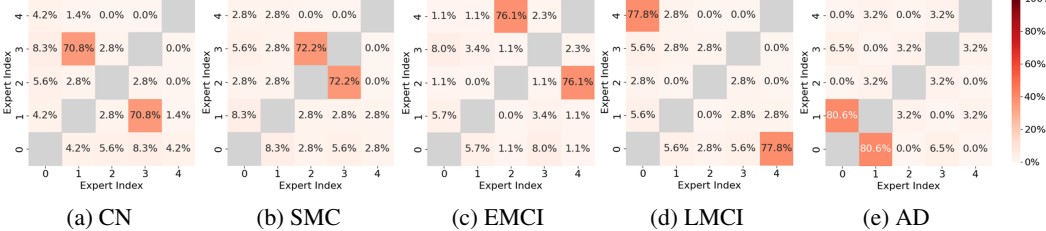

Figure 4: Expert selection patterns across diagnostic groups in ADNI-CT. Each heatmap illustrates the frequency of expert activation for node-level representations, with five experts and $k = 2$.

## 6 CONCLUSION

We proposed a novel framework that extends the spectral graph wavelet to simplicial complexes, enabling multi-scale spectral analysis of both node and edge features in a graph, while operating in a graph-wise adaptive manner. Leveraging the Hodge Laplacian, SSWT adaptively filters graph signals on both nodes and edges, and SpMoE extends it to a multi-resolution learning, enhancing the spectral representation of complex structures. Extensive experiments on ADNI and PPMI datasets show that our model achieves state-of-the-art prediction performance with clinically meaningful interpretation, demonstrating its potential for brain network analysis in neurodegeneration as well as broader applications in both neuroscience and graph analysis.

REPRODUCIBILITY

To ensure the reproducibility of our work, we provide comprehensive implementation details, including model architectures, optimization settings, and baseline tuning protocols, in Sec. D. The datasets used in our experiments, ADNI and PPMI, are publicly available, and the full preprocessing pipeline is described in Sec. C. For the theoretical component, all mathematical assumptions and proofs are explicitly provided in Sec. A and Sec. B. All experiments are conducted with 5-fold cross-validation, and results are reported as mean values with standard deviations. We will also release the full code and setup to facilitate reproducibility of our work.

ETHICS

This study uses only publicly available neuroimaging datasets, ADNI and PPMI, which are distributed for research use. No additional data involving human subjects were collected or used. The datasets are widely adopted benchmarks in the neuroimaging community and contain no personally identifiable information, minimizing privacy concerns.

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

# APPENDIX

In the appendix, we provide supplementary materials to support the main manuscript. Specifically, we include: **1)** a proof of Lemma 1 on the orientation equivariance of our edge filtering; **2)** a detailed derivation of the gating mechanism and the role of noise in expert selection; **3)** detailed description of the ADNI and PPMI datasets used in our experiments; **4)** full implementation details and hyperparameter settings for our model, along with a comprehensive baseline tuning protocols; **5)** an ablation study on the number of experts and top-$k$ selection in SpMoE; **6)** the effectiveness of load balancing losses on model performance; **7)** a discussion on model complexity; and **8)** a discussion of the limitations and broader societal impacts of the proposed method.

## A   PROOF OF LEMMA 1

**Lemma 1.** *Consider a boundary matrix $B_1 \in \mathbb{R}^{N \times M}$ with arbitrary orientations and edge signal $\omega$ of a graph $\mathcal{G}$. Let $P \in \mathbb{R}^{M \times M}$ be an arbitrary diagonal matrix encoding changes in edge orientation, where $P[i,i] = -1$ if the orientation of the $i$-th edge is flipped, and $P[i,i] = 1$ otherwise. Define the reoriented graph $\mathcal{G}'$, with $\omega' = P\omega$ and $B_1' = B_1 P$. Then, the filtered edge signal $\omega_t'$ on $\mathcal{G}'$ satisfies $\omega_t' = P\omega_t$, demonstrating that the ISWT is equivariant to edge orientation.*

*Proof.* Let $P \in \mathbb{R}^{M \times M}$ be a diagonal matrix defined as:

$$P[i,i] = \begin{cases} -1, & \text{if the } i\text{-th edge is flipped,} \\ 1, & \text{otherwise.} \end{cases} \tag{1}$$

Altering the orientation of edges corresponds to flipping the signs of the corresponding columns in $B_1$ as $B_1' = B_1 P$. The modified Hodge Laplacian $\mathcal{L}_1'$ follows as:

$$\mathcal{L}_1' = B_1'^T B_1' = (B_1 P)^T (B_1 P) = P^T B_1^T B_1 P = P^T \mathcal{L}_1 P. \tag{2}$$

Since $P$ is diagonal with entries $\pm 1$, it satisfies $P = P^T = P^{-1}$. Thus, the relationship between $\mathcal{L}_1'$ and $\mathcal{L}_1$ represents a similarity transformation. This implies that they share the same eigenvalues $\Lambda_1$. Let the eigendecomposition of the original Laplacian be $\mathcal{L}_1 = U_1 \Lambda_1 U_1^T$. The eigenvectors of the reoriented graph, denoted by $U_1'$, are related to $U_1$ as follows:

$$\mathcal{L}_1'(PU_1) = (P\mathcal{L}_1 P^{-1})PU_1 = P\mathcal{L}_1 U_1 = PU_1 \Lambda_1. \tag{3}$$

From Eq. (3), we identify that $U_1' = PU_1$. Since $P^{-1} = P$, it also holds that $U_1 = PU_1'$.

Now, consider the wavelet coefficients $W_\omega(t) = \phi_t \cdot \omega = U_1 \mathcal{K}(t\Lambda_1)U_1^T \omega$. Applying the transformation $U_1' = PU_1$ and $\omega' = P\omega$, the coefficients for the reoriented graph are:

$$W_\omega'(t) = U_1' \mathcal{K}(t\Lambda_1)U_1'^T \omega' \tag{4}$$

$$= (PU_1)\mathcal{K}(t\Lambda_1)(PU_1)^T(P\omega) \tag{5}$$

$$= PU_1 \mathcal{K}(t\Lambda_1)U_1^T P^T P\omega \tag{6}$$

$$= PU_1 \mathcal{K}(t\Lambda_1)U_1^T \omega = PW_\omega(t). \tag{7}$$

Similarly, the filtered edge signal $\omega_t = U_1 \mathcal{K}^2(t\Lambda_1)U_1^T \omega$ transforms as:

$$\omega_t' = U_1' \mathcal{K}^2(t\Lambda_1)U_1'^T \omega' \tag{8}$$

$$= (PU_1)\mathcal{K}^2(t\Lambda_1)(PU_1)^T(P\omega) \tag{9}$$

$$= PU_1 \mathcal{K}^2(t\Lambda_1)U_1^T P^T P\omega \tag{10}$$

$$= PU_1 \mathcal{K}^2(t\Lambda_1)U_1^T \omega = P\omega_t. \tag{11}$$

This demonstrates that flipping the orientation of edges only changes the sign of the corresponding components of $W_\omega(t)$ and $\omega_t$, while preserving their magnitudes. This proves orientation equivariance. $\square$

# B  DERIVATION OF LOAD LOSS AND THE ROLE OF $\theta_{\text{NOISE}}$ IN EXPERT SELECTION

As described in Sec. 4.3, the SpMoE layer employs a noisy top-$k$ gating mechanism to dynamically assign input signals to a subset of experts based on their gating scores. To determine expert assignment, we define the gating function as follows:

$$G(x) = \text{Softmax}(\text{TopK}(H(x), k)), \tag{12}$$

where the raw gating score for each $i$-th expert $H(x)_i$ is computed as

$$H(x)_i = (x \cdot \theta_g)_i + \epsilon \cdot \text{Softplus}((x \cdot \theta_{\text{noise}})_i). \tag{13}$$

Here, $(x \cdot \theta_g)_i$ represents the deterministic raw gating score, while $\epsilon \sim \mathcal{N}(0, 1)$ is Gaussian noise introduced to add stochasticity to the selection process. The noise magnitude is controlled by a learnable parameter $\theta_{\text{noise}}$, which is transformed via Softplus function to ensure a non-negative scale.

An expert is activated if its gating score is among the top-$k$ highest scores for the given input $x$. This condition is expressed as:

$$H(x)_i \geq \text{threshold}_k(H(x)), \tag{14}$$

where $\text{threshold}_k(H(x))$ represents the $k$-th largest gating score among all experts for the input $x$. Thus, the probability of selecting expert $i$ is given by:

$$\mathbb{P}(G(x)_i \neq 0) = P(H(x)_i \geq \text{threshold}_k(H(x))). \tag{15}$$

Substituting the definition of $H(x)_i$:

$$P\left((x \cdot \theta_g)_i + \epsilon \cdot \text{Softplus}((x \cdot \theta_{\text{noise}})_i) \geq \text{threshold}_k(H(x))\right). \tag{16}$$

Rearranging,

$$P\left(\epsilon \geq \frac{\text{threshold}_k(H(x)) - (x \cdot \theta_g)_i}{\text{Softplus}((x \cdot \theta_{\text{noise}})_i)}\right). \tag{17}$$

Since $\epsilon \sim \mathcal{N}(0, 1)$, we use the cumulative distribution function (CDF) of the standard normal distribution, $\Phi(\cdot)$, to express this probability as:

$$\mathbb{P}(G(x)_i \neq 0) = \Phi\left(\frac{(x \cdot \theta_g)_i - \text{threshold}_k(H(x))}{\text{Softplus}((x \cdot \theta_{\text{noise}})_i)}\right). \tag{18}$$

Without the noise term $\epsilon$, the gating function becomes deterministic, meaning that expert $i$ is selected if and only if:

$$G(x)_i \neq 0 \quad \text{if and only if} \quad (x \cdot \theta_g)_i \geq \text{threshold}_k(H(x)). \tag{19}$$

In this case, the selection probability reduces to a binary indicator function, which can still achieve balanced expert utilization through load balancing losses. However, introducing noise further enhances balancing by mitigating early-stage expert dominance, facilitating smoother selection transitions, and encouraging more diverse expert utilization throughout training. The stochastic nature of noise also promotes exploration, allowing underutilized experts to receive gradients more consistently and contributing to a more stable and effective learning process.

## C  DETAILED DATASET DESCRIPTION

We utilize two neurodegenerative brain network datasets: the Alzheimer's Disease Neuroimaging Initiative (ADNI) and the Parkinson's Progression Markers Initiative (PPMI).

### C.1  ADNI

The ADNI study (Mueller et al., 2005) is a publicly available dataset designed for Alzheimer's Disease (AD) research, providing multimodal imaging and biomarker data across various stages of cognitive decline. Following prior studies (Choi et al., 2022; Sim et al., 2024; Cho et al., 2024), the structural brain graphs in the ADNI dataset are constructed through a multi-step imaging pipeline. First, skull stripping and tissue segmentation are performed on T1-weighted MRI scans. The brain is then parcellated into 160 anatomical regions of interest (ROIs) based on the Destrieux atlas (Destrieux et al., 2010). Diffusion tensors are estimated from DWI data, and cortical surfaces are reconstructed using FreeSurfer (Fischl, 2012). Probabilistic tractography is applied to extract white matter fiber tracts between ROIs, from which the edges of the graph are defined and weighted by tract counts, reflecting the integrity of structural connectivity (Ciccarelli et al., 2006). Finally, region-wise imaging features are extracted: cortical thickness (CT) from MRI and standardized uptake value ratios (SUVRs) from FDG-PET, where the cerebellum is used as the reference region for SUVR normalization. These modalities are widely used in Alzheimer's Disease (AD) research, with cortical thickness capturing gray matter atrophy and FDG-PET reflecting metabolic deficits (Vemuri & Jack, 2010).

The dataset consists of five diagnostic groups: Cognitively Normal (CN), Significant Memory Concern (SMC), Early Mild Cognitive Impairment (EMCI), Late Mild Cognitive Impairment (LMCI), and Alzheimer's Disease (AD). The demographic distribution of the ADNI dataset is summarized in Tab. 1. It includes a total of 1,323 subjects, with 359 CN, 181 SMC, 437 EMCI, 180 LMCI, and 166 AD cases. The average age within these groups ranges from 70.9 to 74.8 years, with standard deviations between 1.4 and 8.7 years. Gender distribution varies across groups, with CN comprising 178 males and 181 females, while AD includes 102 males and 64 females.

Table 1: Demographics of the ADNI dataset.

| Biomarker | Category | CN | SMC | EMCI | LMCI | AD |
|---|---|---|---|---|---|---|
| Cortical Thickness & FDG | # of subjects | 359 | 181 | 437 | 180 | 166 |
| | Gender (M / F) | 178 / 181 | 69 / 112 | 249 / 188 | 119 / 61 | 102 / 64 |
| | Age (Mean±Std) | 72.8±1.4 | 72.0±5.2 | 71.0±7.9 | 70.9±6.1 | 74.8±8.7 |

### C.2  PPMI

The PPMI dataset (Marek et al., 2011) focuses on understanding the progression of Parkinson's Disease (PD) through imaging and biomarker analysis. Functional brain networks are constructed by parcellating resting-state fMRI into 116 regions based on the AAL atlas (Tzourio-Mazoyer et al., 2002). Node features are extracted from Blood-Oxygen-Level-Dependent (BOLD) signals capturing changes in blood flow, while edge features capture functional connectivity using correlation (Xu et al., 2023).

The dataset consists of three diagnostic groups: Cognitively Normal (CN), Prodromal, and Parkinson's Disease (PD), representing different stages of disease progression. The SWEDD group (Scans Without Evidence for Dopaminergic Deficit) was excluded from our study. Although SWEDD patients present parkinsonian motor symptoms, they show normal presynaptic dopaminergic imaging findings, making them pathophysiologically distinct from PD. Including them could introduce confounding effects by acting as outliers and hindering the model's ability to capture genuine disease progression signals. The official PPMI Data User Guide[1] also describes SWEDD as a small legacy cohort that researchers may wish to exclude depending on their research purpose, which aligns with our decision. The demographic details of the PPMI dataset are presented in Tab. 2. It includes a total

---

[1]https://www.ppmi-info.org/sites/default/files/docs/PPMI%20Data%20User%20Guide.pdf

of 195 subjects, with 15 CN, 67 Prodromal, and 113 PD cases. The average age within these groups ranges from 62.0 to 64.3 years, with standard deviations up to 10.0 years. Gender distribution also varies, with CN consisting of 12 males and 3 females, while the PD group comprises 77 males and 36 females.

Table 2: Demographics of the PPMI dataset.

| Biomarker | Category | CN | Prodromal | PD |
|---|---|---|---|---|
| | # of subjects | 15 | 67 | 113 |
| BOLD | Gender (M / F) | 12 / 3 | 38 / 29 | 77 / 36 |
| | Age (Mean±Std) | 62.0±10.0 | 64.3±8.7 | 64.0±9.5 |

## D IMPLEMENTATION DETAILS

All models, including our model and baseline methods, were implemented using PyTorch and trained on a single NVIDIA RTX 6000 Ada Generation GPU. We used the Adam optimizer and trained each model for 5000 epochs. A grid search was conducted over the following common hyperparameter ranges: the number of hidden units in $\{2, 4, 6, 8, 16, 32, 64\}$, dropout rates in $\{0.2, 0.4, 0.6, 0.8\}$, and learning rates in $\{0.1, 0.01, 0.001, 0.0001\}$.

**Our Settings.** Tab. 3 summarizes the hyperparameters used for our model across datasets. The batch size was set to include all samples in each dataset. Dropout rates were set to 0.6 for ADNI-CT and ADNI-FDG, and 0.4 for PPMI. The learning rates were set to $1 \times 10^{-3}$ for ADNI-CT and ADNI-FDG, and $2 \times 10^{-2}$ for PPMI. The weight for $\alpha$ in Eq. (19) was set to 5 across all datasets. The hidden dimension $h$ was set to 6 for ADNI-CT, 4 for ADNI-FDG, and 16 for PPMI, and we used 2 readout layers across all datasets. The number of node experts $P$ and edge experts $Q$ for SpMoE was set to (5, 5) for ADNI-CT, (5, 4) for ADNI-FDG, and (5, 5) for PPMI. We selected $k = 2$ experts per sample, and all expert scale parameters were initialized to 1.

Table 3: Hyperparameters of our model for experiments.

| Hyperparameter | ADNI-CT | ADNI-FDG | PPMI |
|---|---|---|---|
| Optimizer | Adam | Adam | Adam |
| Number of epochs | 5000 | 5000 | 5000 |
| Weight decay | $5 \times 10^{-4}$ | $5 \times 10^{-4}$ | $5 \times 10^{-4}$ |
| Batch size | 1058 | 1058 | 156 |
| Dropout rate | 0.6 | 0.6 | 0.4 |
| Learning rate | $1 \times 10^{-3}$ | $1 \times 10^{-3}$ | $2 \times 10^{-2}$ |
| Weight for $\alpha$ (in Eq. (19)) | 5 | 5 | 5 |
| Hidden dimension $h$ | 6 | 4 | 16 |
| Number of $F(\cdot)$ layers | 2 | 2 | 2 |
| Number of experts $(P, Q)$ | (5, 5) | (5, 4) | (5, 5) |
| $k$ for top-$k$ selection | 2 | 2 | 2 |
| Initialization of scales | 1 | 1 | 1 |

**Baseline-specific Settings.** To ensure fair and reliable comparisons, we used official open-source implementation for each method. In addition to the shared grid search described above, we further tuned model-specific key hyperparameters to adapt each method to our domain-specific datasets:

- **GDC** (Gasteiger et al., 2019): We evaluated both the Heat Kernel and Personalized PageRank (PPR) variants. For each, we tuned the sparsification threshold $\epsilon \in \{10^{-2}, 10^{-3}, 10^{-4}\}$ for edge weight threshold applied to the diffusion matrix, the top-$k$ values $k \in \{32, 64, 128\}$ for number of neighbors retained per node after diffusion, heat diffusion time $t \in \{1, \ldots, 6\}$, and teleport probability $\alpha \in \{0.05, 0.10, 0.15, 0.20\}$.

- **GraphHeat** (Xu et al., 2019) and **ADC** (Zhao et al., 2021): We tuned the heat diffusion time in the range $t \in \{1, \ldots, 6\}$. For ADC, we also tuned the truncation order of the Taylor expansion $K \in \{5, 10, 15, 20\}$.
- **BrainGNN** (Li et al., 2021): We adjusted the pooling ratio for R-pool layer $\{0.1, 0.2, 0.3, 0.4, 0.5\}$ and the number of ROI communities $K^{(l)} \in \{2, 4, 6, 8\}$.
- **BrainNetTF** (Kan et al., 2022): We tuned the number of transformer heads $\{2, 4, 8\}$ and the number of clustering centers $K \in \{2, 4, 8, 10, 20\}$ in the OCREAD readout module.
- **ALTER** (Yu et al., 2024): We tuned the number of random walk hops $K \in \{2, 4, 8, 16, 32\}$ for the adaptive long-range encoding module. For the transformer encoder, we searched over the number of layers $L \in \{1, 2, 3, 4\}$ and the number of attention heads $M \in \{1, 2, 3, 4\}$.
- **ContrastPool** (Xu et al., 2024): We tuned the trade-off hyperparameters $\lambda_1 \in \{10, 1, 0.1\}$ and $\lambda_2 \in \{10^{-2}, 10^{-3}, 10^{-4}\}$ for the entropy regularization losses on the assignment matrix and the contrast graph, respectively. We also searched over the pooling ratio $\in \{0.3, 0.4, 0.5, 0.6\}$ and the number of pooling layers $L \in \{2, 3, 4\}$.

For each baseline, the best configuration was selected based on validation performance. These settings follow the experimental protocol of AGT (Cho et al., 2024), which is a prior method using the same datasets. Full code and hyperparameter configurations will be released upon acceptance to ensure reproducibility.

# E  ADDITIONAL EXPERIMENTAL RESULTS

## E.1  ABLATION STUDY ON EXPERT NUMBER AND TOP-$k$ SELECTION IN SPMOE

We investigate the impact of varying the number of experts and top-$k$ selection in SpMoE, as shown in Tab. 4 and Tab. 5. Experiments follow the same evaluation protocol as Tab. 1 on the ADNI dataset.

Increasing the number of node and edge experts enhances spectral representation by leveraging a more diverse set of wavelet scales. This improves the model's ability to capture both localized and global connectivity patterns across multiple resolutions. However, an excessive number of experts does not always yield better performance, as seen in Tab. 5, where accuracy saturates or slightly decreases when $P, Q = 7$. This is due to increased redundancy in spectral representation, where multiple experts learn overlapping frequency components rather than capturing distinct spectral features.

The top-$k$ value, which controls the number of active experts per instance, also affects adaptability. A lower value may restrict scale diversity, whereas a higher value can introduce unnecessary complexity. An optimal top-$k$ ensures the model focuses on the most relevant spectral scales, balancing efficiency and expressiveness.

Table 4: Ablation study on expert number and top-$k$ selection in SpMoE for the ANDI-CT.

| Number of Experts | | top-$k$ | Accuracy (%) | Precision (%) | Recall (%) |
|---|---|---|---|---|---|
| $P$ | $Q$ | $k$ | | | |
| 1 | 1 | 1 | $92.5 \pm 1.1$ | $92.0 \pm 1.1$ | $93.4 \pm 1.0$ |
| 3 | 3 | 1 | $93.3 \pm 1.4$ | $93.1 \pm 1.6$ | $93.5 \pm 1.4$ |
| 3 | 3 | 2 | $94.0 \pm 1.2$ | $93.8 \pm 1.3$ | $95.0 \pm 1.0$ |
| 3 | 3 | 3 | $93.9 \pm 1.0$ | $93.8 \pm 0.9$ | $95.0 \pm 0.8$ |
| 5 | 5 | 1 | $94.1 \pm 1.8$ | $93.7 \pm 2.2$ | $94.8 \pm 1.2$ |
| 5 | 5 | 2 | $\mathbf{94.9 \pm 1.3}$ | $\mathbf{94.4 \pm 1.7}$ | $\mathbf{95.4 \pm 0.8}$ |
| 5 | 5 | 5 | $94.3 \pm 1.3$ | $94.0 \pm 1.3$ | $95.2 \pm 1.1$ |

Table 5: Ablation study on expert number and top-$k$ selection in SpMoE for the ANDI-FDG.

| Number of Experts | | top-$k$ | Accuracy (%) | Precision (%) | Recall (%) |
|---|---|---|---|---|---|
| $P$ | $Q$ | $k$ | | | |
| 1 | 1 | 1 | $92.8 \pm 0.7$ | $93.3 \pm 1.3$ | $93.1 \pm 0.9$ |
| 3 | 3 | 1 | $93.2 \pm 1.1$ | $91.7 \pm 1.1$ | $94.2 \pm 1.0$ |
| 3 | 3 | 2 | $95.5 \pm 0.9$ | $95.1 \pm 1.5$ | $95.6 \pm 0.8$ |
| 3 | 3 | 3 | $95.1 \pm 0.9$ | $94.9 \pm 1.0$ | $96.9 \pm 1.2$ |
| 4 | 5 | 2 | $96.2 \pm 1.2$ | $96.4 \pm 1.3$ | $96.1 \pm 1.2$ |
| 5 | 4 | 2 | $\mathbf{96.4 \pm 0.9}$ | $\mathbf{96.5 \pm 1.1}$ | $\mathbf{97.1 \pm 1.1}$ |
| 5 | 5 | 1 | $94.4 \pm 1.5$ | $93.9 \pm 1.3$ | $94.6 \pm 1.2$ |
| 5 | 5 | 2 | $96.3 \pm 0.7$ | $96.2 \pm 1.0$ | $96.9 \pm 0.9$ |
| 5 | 5 | 3 | $96.1 \pm 0.9$ | $95.9 \pm 0.7$ | $97.0 \pm 0.9$ |
| 5 | 5 | 4 | $95.8 \pm 1.1$ | $95.6 \pm 1.3$ | $96.8 \pm 1.1$ |
| 5 | 5 | 5 | $95.5 \pm 1.2$ | $94.8 \pm 0.9$ | $95.8 \pm 1.0$ |
| 7 | 7 | 1 | $94.1 \pm 1.7$ | $93.7 \pm 0.9$ | $94.8 \pm 1.3$ |
| 7 | 7 | 2 | $96.2 \pm 0.7$ | $96.1 \pm 1.1$ | $97.0 \pm 1.2$ |
| 7 | 7 | 5 | $95.9 \pm 1.3$ | $96.1 \pm 1.3$ | $95.7 \pm 0.8$ |

### E.2 ABLATION STUDY ON LOAD BALANCING LOSSES

To evaluate the impact of load balancing losses, we conduct an ablation study comparing model accuracy with and without load balancing losses. The results are summarized in Tab. 6. With $L_{\text{load}} + L_{\text{imp}}$, the model shows consistent improvements across all datasets, with an average accuracy gain of $\sim 1.93\%$p.

This performance boost highlights the importance of balancing expert assignments in the SpMoE framework. Without these losses, the model may suffer from uneven expert utilization during training, where a subset of experts dominates. By ensuring a more even distribution of computational load across experts, load balancing enhances model stability and improves generalization.

Table 6: Comparison of accuracy with and without load balancing losses.

| $L_{\text{ce}}$ | $L_{\text{load}} + L_{\text{imp}}$ | ADNI-CT | ADNI-FDG | PPMI |
|---|---|---|---|---|
| ✓ | | $93.1 \pm 1.9$ | $94.1 \pm 2.3$ | $85.7 \pm 2.9$ |
| ✓ | ✓ | $\mathbf{94.9 \pm 1.3}$ (**+1.8**) | $\mathbf{96.4 \pm 0.9}$ (**+2.3**) | $\mathbf{87.4 \pm 2.1}$ (**+1.7**) |

### E.3 DISCUSSION ON MODEL COMPLEXITY

Regarding the computational cost of spectral decomposition, which constitutes the major complexity of the SSWT, we note that the eigen-decomposition of the Hodge Laplacians $\mathcal{L}_0$ and $\mathcal{L}_1$ incurs a theoretical complexity of $\mathcal{O}(N^3)$ and $\mathcal{O}(M^3)$, respectively, where $N$ and $M$ denote the number of nodes and edges. Nevertheless, this operation is performed only once as a preprocessing step prior to training, which is a common practice in spectral methods. To evaluate the average runtime (over 10 epochs) and memory efficiency of our model, we compared it against several representative baselines, as summarized in Tab. 7. All experiments were conducted on an NVIDIA RTX 6000 Ada GPU, using a batch size that includes the entire ADNI-CT dataset.

Our model exhibits competitive efficiency compared to traditional GNN-based approaches (e.g., GAT) and achieves a substantial reduction in training time and memory consumption over recent transformer-based brain models (e.g., BrainNetTF). Notably, when comparing our sparse expert selection mechanism with a dense variant that utilizes all experts without top-$k$ selection, the sparse version reduces runtime by 49% and memory usage by 24%.

Table 7: Comparison of average runtime (sec/epoch) and memory usage (MB) across models.

| Model | sec/epoch | memory (MB) |
|---|---|---|
| GCN | 0.0091 | 2,147 |
| GAT | 0.0915 | 10,657 |
| BrainNetTF | 0.1634 | 32,243 |
| Ours w/o sparse MoE | 0.1992 | 9,051 |
| Ours | 0.1012 | 6,861 |

## F    DISCUSSIONS

**Limitations.** Although our model demonstrates robust performance across datasets when multiple experts are activated (i.e., $k \geq 2$), as shown in Tab. 4 and Tab. 5, the number of experts and the top-$k$ value must still be carefully chosen to ensure optimal performance. Furthermore, the multi-resolution learning process consumes more memory resources than conventional models. However, due to the sparse activation of experts, our model avoids unnecessary computation, offering improved efficiency over dense expert model as in Tab. 7.

**Broader Impact.** This work presents a novel approach for analyzing brain connectomes, aiming to support the diagnosis of neurodegenerative diseases. The proposed method has the potential to positively impact society by enabling earlier and more accurate detection of conditions such as Alzheimer's and Parkinson's disease, leading to improved patient quality of life and reduced healthcare costs. However, overreliance on model predictions without clinical oversight may lead to misdiagnosis. To avoid this, the model should be used as a decision-support tool alongside expert interpretation within standard diagnostic workflows. These risks can be mitigated by incorporating expert interpretation when using the model in clinical workflows.

**LLMs Usage.** We used Large Language Models (LLMs) solely to aid in polishing the writing of this manuscript. No LLMs were used for generating research ideas, experimental design, data analysis, or any other scientific contributions.

