# OpenReview forum: "Mixture of Spectral Wavelets on Simplicial Complex: Analysis of Brain Connectome with Neurodegeneration"
_ICLR.cc/2026/Conference — ICLR 2026 Conference Withdrawn Submission_

### Official Review · Reviewer_jJeF · 2025-10-24

**Soundness:** 2
**Presentation:** 2
**Contribution:** 2
**Rating:** 2
**Confidence:** 4

**Summary:**

This paper proposes a novel framework, SSWT-SpMoE (Mixture of Spectral Wavelets on Simplicial Complex with Spectral Mixture of Experts), for brain connectome analysis in the diagnosis of neurodegenerative diseases such as Alzheimer’s disease (AD) and Parkinson’s disease (PD). The authors aim to overcome two main limitations of existing Graph Neural Networks (GNNs) in this domain: (1) the neglect of crucial edge-level information due to a predominant focus on node features, and (2) the restricted ability of spectral GNNs to capture diverse frequency components because of fixed-bandwidth filtering. To address these issues, the proposed framework integrates two key components. The Spectral Simplicial Wavelet Transform (SSWT) extends spectral graph wavelet transforms to higher-order simplicial complexes using the Hodge Laplacian, enabling joint multi-scale analysis of both node and edge features. The Spectral Mixture of Experts (SpMoE) employs a gating mechanism to dynamically select the most relevant wavelet scales (“experts”) for each graph, allowing adaptive multi-scale spectral filtering. Together, these innovations enable richer spectral representations and adaptive frequency analysis tailored to each brain graph.

**Strengths:**

### **Technical innovation: Joint spectral node–edge analysis and adaptive filtering**
The paper presents *strong methodological originality* by introducing the Hodge Laplacian into the spectral domain through the Spectral Simplicial Wavelet Transform (SSWT), enabling joint multi-scale spectral analysis of both node and edge features in brain graphs. In addition, the proposed Spectral Mixture of Experts (SpMoE) employs a graph-level gating mechanism to dynamically select relevant spectral wavelet “experts,” effectively overcoming the limitations of fixed-bandwidth filters in conventional spectral GNNs. The effectiveness of these components is further supported by comprehensive ablation studies.

### **Demonstrated generalization across structural and functional networks**
The framework is evaluated on *both structural and functional brain networks*, demonstrating robust and consistent performance across different graph modalities. This cross-modality validation highlights the model’s strong generalization capability and reinforces its applicability to diverse neuroimaging scenarios, reducing concerns of overfitting to a specific data type.

### **Interpretability and clinical relevance**
The interpretability analysis provides compelling evidence of biological plausibility: regions and connections with high Grad-CAM activations align well with established AD/PD-related brain areas such as the thalamus, hippocampus, and putamen. Moreover, the expert selection patterns correspond to different disease stages, suggesting that the proposed model offers not only predictive power but also clinically meaningful and interpretable insights into disease mechanisms.

**Weaknesses:**

### **Unfair experimental setup and limited baselines**
The most critical limitation lies in the fairness of the experimental comparison. The paper mainly compares against models that use BOLD signals as node features, rather than methods that leverage connectivity-matrix-based node features—which have become the de facto standard in brain graph learning. Recent studies, such as ***NeuroGraph: Benchmarks for Graph Machine Learning in Brain Connectomics (NeurIPS 2023)*** , have demonstrated that using connectivity matrix rows as node features yields significantly stronger performance than using raw BOLD signals. Without comparisons to such message-passing GNNs using connectivity-based features, the empirical advantage of the proposed spectral approach remains unconvincing.

**Suggested improvement:** The authors should include comparisons with recent message-passing GNNs (e.g., GraphTransformer, BrainGNN, or models evaluated in NeuroGraph) using connectivity-matrix node features to establish a fair and rigorous benchmark.

### **Limited scope of datasets and unclear domain specificity**
Although the paper evaluates the model on neurodegenerative disease datasets, the scope remains narrow. It does not include experiments on widely used brain network benchmarks such as ***ABIDE (autism) or ADHD-200***, which are commonly adopted in brain connectome modeling. This raises concerns about the generalizability of the proposed method beyond neurodegenerative contexts.

**Suggested improvement:** The authors could either (a) extend experiments to additional benchmark datasets to demonstrate broader applicability, or (b) clearly justify why the proposed edge-focused spectral approach is particularly suitable for neurodegenerative diseases and less so for other brain disorders, thereby sharpening the paper’s conceptual focus.

### **Accessibility and clarity of the method section**
The paper’s theoretical development, while rigorous, may be difficult to follow for readers unfamiliar with spectral graph theory and simplicial complexes. The key advantages of the proposed SSWT-SpMoE framework are not sufficiently illustrated in intuitive or visual terms.

**Suggested improvement:** The authors could enhance clarity by adding conceptual diagrams, toy examples, or intuitive explanations that highlight how SSWT enables joint node–edge analysis and why the adaptive spectral filtering improves over fixed-bandwidth methods.

If the authors can provide additional experiments demonstrating that their method outperforms ***SOTA message-passing GNNs using connectivity-based node features on standard benchmarks***, the technical contribution would be much more convincing, and I would be inclined to reconsider my overall assessment of the paper.

**Questions:**

**1. Scalability and approximation**

Given the model’s reliance on spectral decomposition, how does it scale to larger graphs with tens of thousands of nodes or edges? Have the authors explored approximate spectral methods (e.g., Chebyshev polynomial approximation) to reduce computational cost? If so, how would such approximations affect the SpMoE gating mechanism and its ability to select optimal spectral scales?

**2. Kernel function choice**

The kernel function $\mathcal{K}$ used in Eq. (7) is a critical design choice, and Figure 1 suggests the use of the heat kernel $\mathcal{K}_{s}(\lambda)=e^{-s\lambda}$. Which kernel was actually employed in experiments? How sensitive is the method to the kernel choice, and how was its form or scale parameter determined?

**3. Hodge decomposition and interpretability**

Since the Hodge Laplacian decomposes signals into gradient, curl, and harmonic components, does SSWT-SpMoE explicitly or implicitly leverage these subspaces? Could the authors discuss whether different spectral components correspond to meaningful neurobiological or topological patterns (e.g., gradient-like atrophy flows or rotational dysfunctions) in neurodegenerative diseases?

---

> ### Author Response · Authors · 2025-11-28
> **Response to reviewer jJeF (Part 1)**
>
> W1) Unfair experimental setup and limited baselines
>
> A) We would like to clarify that our experiments already include strong and modern message-passing GNN baselines, such as Brain Network Transformer and BrainGNN. Also, the NeuroGraph benchmark primarily evaluates pre-2020 classical GNNs (e.g., GCN, SAGE, GAT, Cheb), none of which represent the recent state of the art in connectome modeling. Therefore, our baseline suite is not weaker; it in fact includes more advanced and domain-specialized models than those used in NeuroGraph.
>
> W2) Limited scope of datasets and unclear domain specificity
>
> A) Our method is motivated by the characteristic progression patterns of neurodegenerative diseases, where both node-level biological signals and edge-level connectivity undergo consistent, progression-linked degradation. This dual alteration directly matches the modeling assumptions of SSWT–SpMoE, which jointly analyzes multi-scale spectral patterns on nodes and edges. In contrast, datasets such as ABIDE (autism) and ADHD-200 involve neurodevelopmental disorders, which do not exhibit consistent or progressive changes in either node morphology or edge connectivity. Numerous studies report substantial heterogeneity and a lack of consistent degeneration trajectories in these conditions. As such, they are not well aligned with the joint node–edge degeneration model that our approach is designed to capture.
> For these reasons, we focused on ADNI and PPMI, where node–edge biomarkers are well established and directly relevant to evaluating the strengths of SSWT–SpMoE.
>
> W3) Accessibility and clarity of the method section
>
> A) Thank you for the helpful suggestion. We agree that additional intuitive illustrations would further improve accessibility. While Figure 1 and Figure 2 already provide initial visualizations of the localized wavelets and the overall pipeline, we will incorporate additional conceptual diagrams and simplified toy examples in the revised version to more clearly convey how SSWT performs joint node–edge analysis and why adaptive spectral filtering offers advantages over fixed-bandwidth approaches.

---

> ### Author Response · Authors · 2025-11-28
> **Response to reviewer jJeF (Part 2)**
>
> Q1) Scalability and approximation
>
> A) As discussed in Section E.3, the eigen-decomposition of $\mathcal{L}_0$ and $\mathcal{L}_1$ has a theoretical complexity of $O(N^3)$  and $O(M^3)$ with $N$ nodes and $M$ edges, respectively. However, this decomposition is performed only once as a preprocessing step prior to training, which is a common practice in spectral methods. To better ensure scalability to larger graphs, as suggested by (Hammond et al., 2011), polynomial approximation, e.g., Chebyshev, can be adopted to reduce the complexity to linear in the number of non-zero Laplacian entries.
> Regarding the effect of approximate filtering on the SpMoE gating mechanism, as described in Section 4.3, the gating network does not use the spectrally filtered signals as input. Instead, it takes only the original node and edge signals ($f, \omega$) and computes the gating scores to select the optimal scale experts. The gating decision therefore depends only on the graph’s native feature structure and is independent of the specific numerical method used to compute the wavelet filters.
>
> Q2) Kernel function choice
>
> A) As described in the final part of Section 3.2, the low-pass kernel captures information that band-pass kernels cannot, particularly the very low-frequency components near $\lambda=0$. For this reason, both kernels must be used to cover the full spectral range. Accordingly, our model computes both low-pass and band-pass filtered representation (e.g., $f^{low}, f^{mix}$ for nodes) using
> $\mathcal{K}_s^{\text{low}}(\lambda) = e^{-s\lambda}, \quad \mathcal{K}_s(\lambda) = s\lambda e^{-s\lambda}$,
> respectively, and both components are used in all experiments.
> Regarding the kernel choice, these forms satisfy the standard requirements of spectral graph wavelet theory (Hammond et al., 2011) and have clear interpretations. The low-pass kernel is the classical heat kernel, which smooths high-frequency components and captures global structure.
> The band-pass kernel corresponds to the scaled spectral derivative of the heat kernel. This form is chosen for two key reasons:
> Band-pass Property: It acts as a band-pass filter (vanishing at $\lambda=0$ and decaying for large $\lambda$), which allows the model to capture structural transitions and intermediate frequency features that the low-pass filter misses.
> Scale Normalization: The factor $s$ ensures that the peak amplitude is normalized across scales. The maximum value occurs at $\lambda = 1/s$ and is always $1/e$ regardless of $s$, which yields balanced multi-scale responses.
>
> Q3) Hodge decomposition and interpretability
>
> A) Our method does not explicitly rely on Hodge decomposition, i.e., it does not decompose signals into gradient, curl, and harmonic components.  Instead, it operates directly on the spectral decomposition of $L_0$​ and $L_1$​ and performs multi-scale filtering across all eigenmodes. This avoids imposing a fixed gradient/harmonic partition and enables the model to learn task-relevant spectral patterns directly from the data (the curl subspace is absent in graphs, as a graph is a 1-dimensional simplicial complex containing only 0- and 1-simplices).

---

### Official Review · Reviewer_efrH · 2025-10-31

**Soundness:** 4
**Presentation:** 3
**Contribution:** 3
**Rating:** 6
**Confidence:** 4

**Summary:**

This paper develops the wavelet transform on 0-simplices and 1-simplices based on Hodge Laplacian, to overcome the difficulty of jointly analyzing the node and edge features. Adaptive wavelet scale selection is applied in the node and edge representation learning based on $L_0$ and $L_1$ operators. Application in brain network datasets validates the effectiveness of the proposed model.

**Strengths:**

1) The Hodge Laplacian adopted in this paper solves the mixture of node feature and edge feature in traditional GNNs. It allows the representation learning of node feature and edge feature within its own domain, which solves the difficulty of processing node and edge feature simultaneously.

2) The network proposed in this paper adopts the wavelet scale selection in multi-scale representation learning, which introduces flexibility compared with the learning with fixed wavelet.

3) The experiment part is adequate with multiple datasets considered and results visualization provided, which is convincing and straightforward.

**Weaknesses:**

1) The method proposed in this paper requires a complete eigenvalue decomposition of both the $L_0$ and $L_1$ operators, which leads to huge computational and storage costs, since the complexity of EVD is O($N^3$) and the size of eigenvalue matrix is O($N^2$).

2) The representation learning strucure of the proposed method is a shallow network with adaptive graph wavelet transform, and lacks a hierachical structure. Meanwhile, the proposed method is sensitive to the hyper-parameters, such as the range of wavelet scales and the number of experts.

**Questions:**

1) In this paper, the node feature and edge feature are processed within its own space, and only interacts with each other in the final step of label prediction. Will that lead to a potential mismatching of the node and edge features?

2) It seems that the scale selection mainly concentrate on a certain scale. Will more experts lead to a better performance?

---

> ### Author Response · Authors · 2025-11-28
>
> W1)
> As described in Sec.E.3 of the supplementary material, the eigen-decomposition of the Hodge Laplacians are performed only once as a preprocessing step prior to training, which is a common practice in spectral methods.
> For larger-scale graphs, approximate or truncated spectral decompositions (e.g., randomized SVD or Chebyshev polynomial-based approximations) can be readily incorporated into our framework without modifying the overall model formulation.
>
>
> W2)
> The proposed model should not be regarded as shallow in a limiting sense;
> in graph neural networks, stacking layers hierarchically often leads to the well-known oversmoothing problem (e.g., [1], [2]), where deeper architectures cause node and edge representations to converge toward similar values and graph structural distinctions to vanish.
> This issue becomes even more critical in brain networks, where subtle region-wise differences and their connections reflect disease-related alterations, and preserving these distinctions is essential for reliable downstream analysis.
> Regarding the comment on hyper-parameter sensitivity, the proposed method does not exhibit the level of sensitivity suggested in the review. Unlike prior spectral wavelet approaches such as SGWT, which require manually specifying the scale ranges, our method learns the scale parameters directly and relies on the SpMoE gating mechanism to automatically select the appropriate scale sets for each graph. This substantially reduces dependence on manual hyper-parameter choices.
> Moreover, as shown in Tables 4 and 5, once the top-$k$ selection is set to $k \ge 2$,
> the performance remains stable across a broad range of expert numbers,
> and shows only marginal changes beyond a moderate number of experts (e.g., $P=Q=5$).
> These observations demonstrate that the model does not require extensive hyper-parameter tuning and is robust to reasonable variations in scale ranges and expert counts.
>
> Q1) In our setting, processing node and edge features in their respective spaces does not lead to mismatching issues.
> First, node- and edge-level signals capture fundamentally different types of information in brain networks (regional attributes vs.\ connectivity strengths), and treating them within separate simplicial orders is consistent with the Hodge framework.
> Second, both $L_0$- and $L_1$-based wavelet transforms are derived from the same boundary operator $B_1$, which ensures that the spectral bases for nodes and edges are structurally aligned rather than independent.
> Third, the SpMoE gating mechanism jointly learns scale selections for both domains, providing implicit coordination between node- and edge-level representations.
> As a result, the final classifier receives two coherent multi-scale representations, and we did not observe any inconsistency or degradation arising from this design in any experiment.
>
> Q2) Thank you for the question. We already analyzed this effect in detail in Appendix E.1 (Table 4–5). As shown there, increasing the number of experts does not continually improve performance: accuracy saturates and even slightly decreases when too many experts are used due to redundancy in spectral representation. The model naturally concentrates on the most informative scales, and adding more experts beyond this point provides no additional benefit.
>
> [1] Keriven, Nicolas. "Not too little, not too much: a theoretical analysis of graph (over) smoothing." NeurIPS, 2022
>
> [2] Rusch, T. Konstantin, Michael M. Bronstein, and Siddhartha Mishra. "A survey on oversmoothing in graph neural networks." arXiv preprint arXiv:2303.10993, 2023

---

### Official Review · Reviewer_a7hG · 2025-10-31

**Soundness:** 2
**Presentation:** 2
**Contribution:** 3
**Rating:** 4
**Confidence:** 3

**Summary:**

This paper presents a novel framework for brain network analysis that jointly models node and edge features to better understand neurodegenerative disease progression. The method introduces two core components: the Spectral Simplicial Wavelet Transform (SSWT), which enables multi-scale spectral analysis using both node and edge information, and the Spectral Mixture of Experts (SpMoE), which performs adaptive, data-dependent spectral filtering. By capturing richer spectral representations and dynamically selecting relevant frequency scales, the framework overcomes the limitations of fixed-bandwidth spectral GNNs. Experiments on benchmark brain network datasets demonstrate improved classification performance and interpretability, suggesting strong potential for advancing neurodegenerative disease analysis.

**Strengths:**

1. The paper proposes a novel framework extending spectral graph wavelets to simplicial complexes, enabling multi-scale spectral analysis of both node and edge features.

2. The paper demonstrates state-of-the-art performance on ADNI and PPMI datasets, with results supporting the model’s clinical interpretability and applicability to brain network analysis.

**Weaknesses:**

1. The analysis of model behavior appears to focus primarily on the SpMoE component, while the justification and deeper examination of the SSWT design are relatively limited. Providing more analysis or empirical evidence to support the effectiveness and design choices of SSWT would strengthen the paper and clarify its individual contribution within the overall framework.

2. The paper references Hodge-GNN [1], which also utilizes the Hodge Laplacian in its implementation, but it is not discussed in the related work or included as a baseline for comparison.

3. Additional details, such as the data splits and number of classes in the graph classification tasks, should be provided. Some results in the main table differ from those in the original paper, and these details could help clarify the discrepancies.

4. The color maps in Figure 3 could be improved, as the edges are currently too faint to be clearly identified.

[1] Joonhyuk Park, et al. Convolving directed graph edges via Hodge Laplacian for brain network analysis. In MICCAI, 2023.

**Questions:**

Please refer to the weaknesses section.

---

> ### Author Response · Authors · 2025-11-28
>
> W1) SSWT was carefully designed to generalize spectral graph wavelets from nodes to edges, enabling joint multi-scale analysis of connectomes. This design is motivated by the following considerations:
>
>  (1) We adopt the Hodge Laplacian because it naturally extends the graph Laplacian while preserving its key spectral properties (real, symmetric, positive semi-definite), allowing rigorous wavelet construction on both nodes and edges.
>
>  (2) The heat and band-pass kernels were chosen to separately capture global and localized structural variations in brain networks.
>
>  (3) Restricting to 0- and 1-simplices focuses on the most meaningful topological orders (regions and their pairwise connections) for neuroimaging data.
>
> Empirically, Fig. 1 visualizes spatially localized filters of SSWT, confirming effective scale control. Tab. 1 shows that SSWT alone already shows powerful performance, and Tab. 3 demonstrates that combining edge signals further improves accuracy. These results validate that our design choices lead to a principled and effective multi-scale representation.
>
> W2) As Hodge-GNN uses only edge connectivity without node features, while our framework jointly models node and edge signals, a direct comparison would not be appropriate. We will clarify this distinction in the revision.
>
> W3) It is stated that all experiments were conducted with 5-fold cross-validation and the number of classes are 5 for ADNI and 3 for PPMI. Regarding the differences in reported results, there are substantial variations in experimental setup between our work and prior studies, and moreover, our experiments are more robust and clinically meaningful. For instance, our ADNI experiments use 1323 structural connectomes with node features from cortical thickness or FDG-PET over the Destrieux atlas, both established biomarkers of Alzheimer’s disease.
>
> In contrast, ALTER performs binary classification on fMRI data with only 130 samples, and ContrastPool uses fMRI for 6-way classification on the Schaefer atlas with higher class imbalance. For PPMI, we use the AAL atlas, whereas ContrastPool uses Schaefer. Additionally, we re-tuned all baselines for fairness, which may also contribute to the numerical differences.
>
> W4) We appreciate the reviewer’s suggestion. We will update Fig. 3 with improved color maps and enhanced edge visibility to ensure clearer identification of connections.

---

### Official Review · Reviewer_6Ekx · 2025-11-02

**Soundness:** 2
**Presentation:** 2
**Contribution:** 1
**Rating:** 2
**Confidence:** 4

**Summary:**

The work is to propose a neural network framework to analyse data of the brain connectome, with an application to assess neurodegeneration. The core of the work is to develop an adaptive method to use spectral wavelets defined on simplicial complexes (nodes and edges). Signals on the nodes and on the edges are then processed through these wavelets. In this work, a method is designed to be fully end-to-end trainable to learn the optimal scales for each of them; this aspect is obtained by introducing a layer which aggregates the wavelet coefficients obtained at various scales, and estimate a weight (the “gating function”) to be used to mix the wavelet coefficients.

The work is sound but it’s not novel. The originality is clearly overstated: the 1st claimed contribution is: “We extend the Spectral Graph Wavelet Transform (SGWT) to higher order simplicial complex via Hodge Laplacian”. This does not hold: wavelets deigned from then Hodge Laplacian were already proposed in Roddenberry et al., 2022 ; note also that NN for simplicial complexes are also designed in Ebli et al., 2020 (where, even if they are not using wavelets (and band-pass filter), one finds the same construction as the one in eq. as (7), yet for a low-pass filter). Now, the authors account for the work of Yang et al. (2022) who developed simplicial neural networks also.

The 2nd claim is the design of the mixture of experts to select the scales. This is not that original as, the authors tell so, it’s coming from Shazeer et al. (2017) (for the gating functions, for the load balancing loss). This article has been quoted nearly 4000 times ; so incorporating it in a framework is not that original. And I have not seen any originality for this instance.

The 3rd claim is that it works. It seems so on the provided examples. However, they are specific to the domain of neuroscience and from Table 1, the results are incremental as compared to what AGT provides.

References:

Ebli, S., Defferrard, M., Spreemann, G.: Simplicial neural networks. arXiv preprint arXiv:2010.03633 (2020)

Roddenberry, T.M., Frantzen, F., Schaub, M.T., Segarra, S.: Hodgelets: Localized spectral representations of flows on simplicial complexes.  ICASSP 2022, pp. 5922–5926 (2022)

**Strengths:**

The strength of the article is :

1. The overall idea is sound, and the resulting method appears to work for the analysis of data on neurodegenerative illnesses.

**Weaknesses:**

The weakness of the article are numerous, as written in the summary. The claimed contributions are not important, nor actually novelties for the first two. Among weaknesses:

1. **The work is incremental**. Spectral wavelets were already considered (see at least Roddenberry et al., 2022) ; Mixture of experts is not novel ; on the task under question, AGT from 2024 obtains similar results.

2. The work provides no generalisation at all in the domain of ML: it is limited to node and edges (without any insight to higher order simplicial complexes); it fails to capture other higher-order relations (e.g. edge-to-edge relations through triangles); it is tested only on one task on brain dataset for neurogenerative disease. There are many elements of interpretation for people in neuroscience in section 5 which might be too technical for most of the audience (and at least, they are too technical for the reviewer to check their correctness and their relevance; the conference is one in ML, not in neuroscience).

3. Section 4.1, including Lemma 1, is trivial from the existing literature (on graph signal processing with wavelets and with simplicial complexes). Also, the authors states to things which appear wrong to me:

* Section 4.1 considers “a directed graph” ; this is misleading. The graph has to be directed for the use of Hodge Laplacians, so that a direction is set to know when a positive is positive between two nodes, by the features obtained are in truth agnostic to direction. So it’s better to say that one considers an undirected graph, and then that one puts some arbitrary orientation  to account for the flows.

* “This ensures that L1 can be reliably utilized for robust graph analysis in any undirected setting.” ; I think it holds only of one uses the magnitude of the wavelet coefficients; keeping the sign will cause problems, I think.

4. In 4.2, the authors used a sampled version of the continuous wavelet transforms, for a discrete set of scales: this does not constitute a complete representation if done without any care. Shouldn’t the scale be constrained somehow ?

5. In 4.3: one should explicitly define Softmax, Softplus, what is an expert, and so one. The whole paragraph on “Gating Network” is not well explained and not sufficient, and, if one is not familiar with previous works, difficult to follow.

6. For Eq. (13), $f_{s_p}$ and $\omega_{t_q}$ should be defined with mathematical formulas, not only words underneath.

7. The role of the parameter $h$ (which appear to be crucial) is not really explained in 4.3.

8. The choices of the kernels $\mathcal{K}$ appear to be arbitrary; why these ones ? Why not combine band-pass and low-pass (scaling functions) ones as in usual WT ? Is it sound to have the same hidden dimension h for low-pass and band-pass features ?

9. For experiments, a threshold is applied on brain networks, while it is known that many results about brain connectome can be sensitive to thresholding. Did you check the robustness to thresholding ?

10. In 5.2, one expects that adding edge signals give more information ad more leeway (more parameters) to learn specific properties of the data. One would expect to see in the ablation study also: use of Nodes + LP only ; use of Nodes + BP only.

**Questions:**

I already asked my questions above.

---

> ### Author Response · Authors · 2025-11-28
> **Response to reviewer 6Ekx (Part 1)**
>
> W1) We thank the reviewer for highlighting the work of Hodgelets (Roddenberry et al., 2022). We acknowledge that Hodgelets first introduced wavelets on simplicial complexes using the Hodge Laplacian, and will clearly cite and discuss this prior work in the revised manuscript. Nevertheless, our proposed Spectral Simplicial Wavelet Transform (SSWT) offers substantial and conceptually distinct contributions beyond prior work.
>
> Specifically, SSWT (1) jointly models node–edge spectral representations via $L_0$ and $L_1$; (2) introduces trainable multi-scale kernels with a Spectral Mixture of Experts (SpMoE) for adaptive spectral filtering; and (3) formally proves orientation equivariance, ensuring robustness to arbitrary edge orientations. While Hodgelets focuses on analytic wavelet dictionaries for edge flows and leverages Hodge decomposition for flow analysis, our SSWT adopts a spectral decomposition viewpoint and is designed as an end-to-end learnable, adaptive and orientation-robust spectral framework for graph-level prediction on undirected connectomes, where edges encode symmetric connectivity rather than directed physical flow.
>
> Regarding neural networks for simplicial complexes (Ebli et al., 2020; Yang et al., 2022), these models employ polynomial-based low-pass filters over the Hodge Laplacian, whereas our SSWT performs spectral wavelet filtering that jointly combines band-pass and low-pass components for richer multi-scale representations.
>
> Furthermore, while we adopt the general MoE principle from Shazeer et al. (2017), our contribution lies in its novel spectral adaptation—the gating mechanism dynamically selects wavelet-scale experts based on spectral graph structure rather than feature channels.
> Finally, our method consistently outperforms AGT across all datasets (+3.3%p average accuracy), while additionally providing interpretable, multi-scale spectral representations.
>
> W2) We appreciate the reviewer’s thoughtful comments regarding the scope and generality of our work.
>
> **Higher-order generalization.**
>
> We agree that the current manuscript does not explicitly provide insights or experiments on simplices of order r≥2. Our implementation of the SSWT focuses on 0- and 1-simplices (nodes and edges), as brain connectomes are represented as 1-dimensional simplicial complexes, and we restricted our analysis to this structure for clarity. Nevertheless, the SSWT formulation is mathematically general under the r-Laplacian hierarchy $L_r = B_r^T B_r + B_{r+1}B_{r+1}^T​$. In principle, it can be extended to capture higher-order relations—e.g., edge-to-edge interactions through triangles via the upper Laplacian term $B_2B_2^T$​ and triangle-to-triangle interactions through shared edges via $L_2=B_2^T B_2$—by applying the same SSWT formulation to each Hodge Laplacian. Although such extensions were beyond our present scope, we recognize them as an important future direction and will explicitly discuss this generalization potential in the revised manuscript.
>
> **Dataset choice and domain applicability.**
>
> Our proposed SSWT is a general framework applicable to any graph or simplicial data; however, the brain network is one of the few real-world domains that inherently require joint spectral modeling of both node and edge features. Standard graph benchmarks such as TU datasets [1] or OGBN datasets [2] contain edges that represent only simple categorical or binary connectivity, whereas brain connectomes include continuous-valued edge weights and biophysically meaningful node attributes.
>
> Moreover, neurodegenerative disorders such as Alzheimer’s and Parkinson’s provide an ideal validation setting, as they exhibit pathological alterations at both the node and edge levels—for instance, cortical atrophy in Alzheimer’s manifests as node-level degeneration, while white-matter degradation reflects disrupted edge-level connectivity.
>
> We therefore evaluated SSWT on two clinically validated datasets (ADNI and PPMI), which capture these phenomena and demonstrate the model’s effectiveness.
>
> Given that our submission falls under ICLR’s primary area “applications to neuroscience & cognitive science,” we consider this focus not a limitation but a deliberate, scientifically grounded choice illustrating SSWT’s relevance to both ML methodology and neuroscience.
>
> [1] Morris, Christopher, et al. "Tudataset: A collection of benchmark datasets for learning with graphs." arXiv preprint arXiv:2007.08663 (2020).
>
> [2] Hu, Weihua, et al. "Open graph benchmark: Datasets for machine learning on graphs." Advances in neural information processing systems 33 (2020): 22118-22133.

---

> ### Author Response · Authors · 2025-11-28
> **Response to reviewer 6Ekx (Part 2)**
>
> W3) We respectfully disagree with the assertion that Section 4.1, including Lemma 1, is trivial. Our connectomes are undirected, and an arbitrary orientation must be assigned to construct the Hodge Laplacian. In this setting, orientation is a purely algebraic convention rather than a semantic property of the data, so it is important that the learned representations do not depend in an uncontrolled way on this arbitrary choice. Orientation equivariance guarantees exactly this: if edge orientations are changed by a diagonal sign-flip operator $P$, the filtered edge representations transform in a predictable and globally consistent manner, $\omega_t' = P\omega_t​$. In other words, different orientation conventions do not induce arbitrary distortions in the spectral representation, but only a structured sign change that can, in principle, be accounted for. This “control” over how orientation affects the representation is what we intended by “robust graph analysis” in the undirected setting. To the best of our knowledge, such orientation equivariance has not been previously established for Hodge–Laplacian–based wavelet or multi-scale spectral filtering, which motivates our formal proof.
>
> Regarding the wording “a directed graph,” our intention was to distinguish the true directionality present in directed graphs from the purely algebraic orientation required for undirected graphs. In a directed graph, the assigned direction is a semantic property of the data and orientation equivariance is unnecessary. In contrast, in undirected connectomes, edge orientation is arbitrary and must not influence the learned representation—hence the necessity of establishing orientation equivariance.
> We will revise the manuscript to make these distinctions explicit and to avoid any potential ambiguity.
>
> W4) We appreciate the reviewer’s concern about using a sampled version of the continuous wavelet transform. We clarify that our primary goal in this work is to obtain discriminative multi-scale spectral representations for brain network classification, rather than to construct a perfectly reconstructive transform with explicit frame bounds (as in classical SGWT, Hammond et al., 2011).
>
> However, our design is carefully structured to prevent systematic information loss. We always use both a scaling function (low-pass) and wavelet functions (band-pass) at positive scales. The low-pass kernel $e^{-s\lambda}$ emphasizes global, smooth structural information by assigning non-zero response to small eigenvalues, while the band-pass kernels $s\lambda e^{-s\lambda}$ at different scales emphasize intermediate and higher eigenvalues. Consequently, for each Laplacian eigenvalue $\lambda_i \ge 0$, at least one filter yields a non-zero response, preventing any portion of the spectrum from being systematically ignored.
>
> Moreover, the SpMoE enhances spectral coverage in a data-driven manner by dynamically selecting and weighting scale-specific experts based on the input graph structure. This allows the model to adaptively emphasize the frequency bands most relevant to each brain network, offering richer representations than fixed sampling.
> Regarding the constraints on scale parameters $s$ and $t$, while we do not impose formal frame-bound constraints, these scales are trainable and are optimized end-to-end through the classification loss $L_{\text{ce}}$​. This optimization process acts as an implicit constraint by driving the scales toward values that maximize discriminative power for the task. Our empirical results demonstrate that this approach yields effective performance in practice, even without enforcing formal frame tightness.
>
> W5, W6, W7) We thank the reviewer for the valuable comments. While Appendix B already provides a detailed derivation of the gating mechanism, we agree that the main text should be more self-contained. In the revised manuscript, we will clarify the gating workflow and provide a formal definition of an expert around lines 259–260. As for Softmax and Softplus, including their definitions directly in the main text would indeed improve readability, but given the space constraints, we will instead add their formal definitions to Appendix B. For Eq. (13), we will explicitly define $f_{s_p}$ and $\omega_{t_q}$ in mathematical form. Regarding the parameter $h$, it simply denotes the embedding dimension of the intermediate representation produced by the SpMoE layer. In practice, h controls the representation capacity of the linear projections $\theta_0$​ and $\theta_1​$, and is treated as a standard hyperparameter. The specific values chosen for each dataset are provided in Appendix D.

---

> ### Author Response · Authors · 2025-11-28
> **Response to reviewer 6Ekx (Part 3)**
>
> W8) We appreciate the reviewer’s comment regarding the choice of kernels and architectural design. Our kernel design is not arbitrary: we chose simple parametric forms that satisfy the standard requirements of spectral graph wavelet theory (Hammond et al., 2011) and have clear interpretations. Specifically, we use:
>
> $\mathcal{K}_s^{\text{low}}(\lambda) = e^{-s\lambda}, \quad \mathcal{K}_s(\lambda) = s\lambda e^{-s\lambda}.$
>
> The low-pass kernel is the classical heat kernel, which smooths high-frequency components and captures global structure.
>
> The band-pass kernel corresponds to the scaled spectral derivative of the heat kernel. This form is chosen for two key reasons:
>
> 1. Band-pass Property: It acts as a band-pass filter (vanishing at $\lambda=0$ and decaying for large $\lambda$), which allows the model to capture structural transitions and intermediate frequency features that the low-pass filter misses.
>
> 2. Scale Normalization: The factor $s$ ensures that the peak amplitude is normalized across scales. The maximum value occurs at $\lambda = 1/s$ and is always $1/e$ regardless of $s$, which yields balanced multi-scale responses.
>
> Regarding design choices, we intentionally deviate from traditional wavelet transforms to prioritize classification over signal reconstruction. We treat the low-pass component as a fundamental structural baseline that must be preserved; therefore, we concatenate it with band-pass features rather than summing them, preventing the risk of the gating mechanism inadvertently filtering out essential global context. Furthermore, we employ the same hidden dimension $h$ and shared projection matrices ($\theta_0$ for nodes and $\theta_1$ for edges) for both components to map them into a unified latent space. This design ensures that global topology (low-pass) and local variations (band-pass) are treated as complementary sources with equivalent representational capacity, enabling the model to learn consistent features from both perspectives without dimensional bias.
> We will clarify these motivations in the revised manuscript.
>
> W9, W10) Thank you for raising these points. Robustness to thresholding and the additional ablations you suggested (Nodes + LP only, Nodes + BP only) are both meaningful extensions. We agree they would strengthen the study and will include them as further experiments in the revised version.

---

### Note · Authors · 2026-04-21

I have read and agree with the venue's withdrawal policy on behalf of myself and my co-authors.

---

### Meta-Review · Area_Chair_HjRR · 2026-01-09

**Summary:**

The four reviewers unanimously identified fundamental weaknesses that led to rejection, with consensus that the paper lacks sufficient novelty, generality, and rigor to meet ICLR’s acceptance bar. They viewed the proposed components (SSWT and SpMoE) as largely incremental extensions of existing spectral wavelets and MoE techniques, with limited originality and overstated novelty claims. Reviewers further criticized the narrow applicability—both methodologically (restricted to nodes and edges, not true higher-order simplices) and empirically (evaluated only on two neurodegenerative datasets without standard benchmarks)—as well as weak theoretical justification, questionable design choices, scalability concerns, and insufficient ablation and baseline comparisons. Combined with marginal empirical gains, domain-specific impact, and poor clarity and presentation, these issues collectively undermined the paper’s contribution and led to a clear rejection decision.

**Reviewer Concerns:**

Issues partially mitigated:

- Originality distinction: The authors listed differences from work such as Hodgelets (joint node-edge, trainability, orientation equivariance), but these differences were still seen as incremental tweaks rather than substantial advances by reviewers.

- Theoretical detail clarification: The explanations of orientation, kernel definition, and gating mechanism were helpful, but no substantial proofs or new analyses were provided.

- Experimental detail supplementation: Data splitting, number of classes, and partial ablation commitments were provided, but these are "remedial" in nature and do not change the fact that the experiments are still limited to two datasets.

- Computational complexity: The potential of preprocessing + Chebyshev was acknowledged, but the paper itself did not realize it, and the promise of "future work" is unconvincing.

Unresolved issues:
- Novelty clarity.

- It merely "discusses" higher-order potential without providing any evidence or preliminary experiments; it fails to address any non-brain connectomics tasks, making it difficult to escape the label of "applied paper in neuro" rather than "ML method., leading to a lack of generality.

- Experimental robustness and fair comparison: the rebuttal fails to offer new results (such as threshold ablation or more baseline re-runs), relying solely on verbal promises; the performance gains are too small to offset the aforementioned deficiencies.

- Broader impact and conference positioning: ICLR prioritizes general ML innovation over domain-specific fine-tuning. Even within the subfield of neuroscience, this hardly qualifies as a state-of-the-art advancement.

**Reviewer Scores:**

Reviewer 6Ekx (Initial 2): This reviewer is the most adamant, deeming the contribution poor. Even with discussion, it's unlikely to exceed 3 (clear reject) because the novelty issue is deeply ingrained.

Reviewer a7hG (Initial 4): The initial score is already marginally below. If the authors provide stronger evidence of ablation and generality, it might rise to borderline accept, but novelty will still be a bottleneck, making a 6 unlikely.

Reviewer efrH (Initial 6): The most lenient reviewer. If more clinical value or robustness is seen in the discussion, it might maintain 6 or slightly decrease to 4. However, given the low scores of the other three reviewers, the overall score is unlikely to improve.

Reviewer jJeF (Initial 2): Similar to 6Ekx, the scope and baseline issues are deeply ingrained.

---

### Decision · Program_Chairs · 2026-01-26

Reject